# The *Chlamydia trachomatis* inclusion membrane protein CT006 associates with lipid droplets in eukaryotic cells

Joana N. Bugalhão[1,2], Maria P. Luís[1,2], Inês S. Pereira[1,2], Maria da Cunha[1,2], Sara V. Pais[2¤], Luís Jaime Mota[1,2]*

1 Associate Laboratory i4HB - Institute for Health and Bioeconomy, NOVA School of Science and Technology, NOVA University Lisbon, Caparica, Portugal, 2 UCIBIO – Applied Molecular Biosciences Unit, Department of Life Sciences, NOVA School of Science and Technology, NOVA University Lisbon, Caparica, Portugal

¤ Current address: Interfaculty Institute of Microbiology and Infection Medicine (IMIT), Faculty of Medicine, University of Tübingen, Tübingen, Germany

* ljmota@fct.unl.pt

**Data Availability Statement:** All relevant data are within the manuscript and its Supporting information files.

## Abstract

*Chlamydia trachomatis* causes genital and ocular infections in humans. This bacterial pathogen multiplies exclusively within host cells in a characteristic vacuole (inclusion) and delivers proteins such as inclusion membrane proteins (Incs) into the host cell. Here, we identified CT006 as a novel *C. trachomatis* protein that when expressed ectopically eukaryotic cells can associate with lipid droplets (LDs). A screen using *Saccharomyces cerevisiae* identified two Incs causing vacuolar protein sorting defects and seven Incs showing tropism for eukaryotic organelles. Ectopic expression in yeast and mammalian cells of genes encoding different fragments of CT006 revealed tropism for the endoplasmic reticulum and LDs. We identified a LD-targeting region within the first 88 amino acid residues of CT006, and positively charged residues important for this targeting. Comparing with the parental wild-type strain, cells infected by a newly generated *C. trachomatis* strain overproducing CT006 with a double hemagglutinin tag showed a slight increase in the area occupied by LDs within the inclusion region. However, we could not correlate this effect with the LD-targeting regions within CT006. We further showed that both the amino and carboxy-terminal regions of CT006, flanking the Inc-characteristic bilobed hydrophobic domain, are exposed to the host cell cytosol during *C. trachomatis* infection, supporting their availability to interact with host cell targets. Altogether, our data suggest that CT006 might participate in the interaction of LDs with *C. trachomatis* inclusions.

## Introduction

Infections by the Gram-negative bacterial pathogen *Chlamydia trachomatis* are a significant public health concern. *C. trachomatis* is the leading cause of sexually transmitted bacterial infections worldwide [1] and is also the causative agent of trachoma, the most common

**Funding:** This work was supported by Fundação para a Ciência e Tecnologia (FCT) through grants PTDC/BIA-MIC/28503/2017 and PTDC/IMI-MIC/1300/2014 attributed to LJM, and in the scope of the projects UIDP/04378/2020 and UIDB/04378/2020 of the Research Unit on Applied Molecular Biosciences – UCIBIO, and LA/P/0140/2020 of the Associate Laboratory Institute for Health and Bioeconomy - i4HB. JNB and SVP were supported by PhD fellowships PD/BD/128214/2016 and PD/BD/52210/2013, respectively, within the scope of the PhD program Molecular Biosciences (PD/00133/2012) funded by FCT. ISP and MPL were supported by PhD fellowships SFRH/BD/129756/2017 and SFRH/BD/144284/2019, also funded by FCT. The funders had no role in study design, data collection and analysis, decision to publish, or preparation of the manuscript.

**Competing interests:** The authors have declared that no competing interests exist.

infectious cause of blindness, which still affects about 1.9 million people in developing countries [2].

*C. trachomatis* is part of the *Chlamydiae* Phylum, comprising obligate intracellular bacteria sharing a biphasic developmental cycle, characterized by the interchange between an extracellular form, the infectious elementary body (EB), and an intracellular form, the non-infectious and replicative, reticulate body (RB). EBs attach to host cells and promote their entry into a membrane bound-vacuolar compartment, known as inclusion. Early after uptake, EBs differentiate into RBs and later in the cycle RBs differentiate asynchronously into EBs, which eventually exit the host cell and can infect neighboring cells [3].

Throughout the developmental cycle, *C. trachomatis* inclusions escape from the normal endo-lysosomal pathway, while interacting with other host cell pathways, including host vesicular and non-vesicular transport, to sustain bacterial survival and replication and also for growth of the inclusion. For instance, *C. trachomatis* inclusions are able to acquire sphingomyelin and cholesterol from the Golgi apparatus and multivesicular bodies; ceramide from the endoplasmic reticulum (ER); and some organelles are translocated intact into the inclusion lumen, such as peroxisomes, possibly as a source of metabolic enzymes, and lipid droplets (LDs) (reviewed in [3]). LDs are ER-derived organelles, composed by a core of neutral lipids surrounded by a monolayer of phospholipids, cholesterol, and several associated proteins. Therefore, they can be targeted by *C. trachomatis* to interfere with lipid homeostasis and as a source of host cell lipids and enzymes [4–10].

To manipulate host cells *C. trachomatis* uses a type III secretion (T3S) system, which allows the delivery of >70 chlamydial effector proteins directly into the host cell cytosol, to the inclusion membrane, and to the inclusion lumen [11, 12]. Among these effectors, there is a group named inclusion membrane proteins (Incs) characterized by at least one bilobed hydrophobic motif mediating their insertion into the membrane of the inclusion [13]. Bioinformatic predictions identified ~60 *C. trachomatis* Incs [14–16] and their localization in the inclusion membrane has been experimentally confirmed for 36 of them [11, 17]. The privileged localization of these proteins anticipated important roles in *C. trachomatis* -host cell interactions, which had been shown for several Incs (reviewed in [11]). For instance, CT229/CpoS inhibits host cell death [18, 19] and interacts with multiple Rab GTPases [20, 21]; CT223/IPAM manipulates host cell microtubules [22]; CT813/InaC also manipulates microtubules and induces actin assembly and Golgi redistribution around the inclusion [23, 24]; and CT116/IncE [25] and CT119/IncA [26] interfere with host cell vesicular trafficking. In addition, a large proteomics study identified several potential interactions of 38 putative Incs with host cell proteins [25]. However, there are still many uncharacterized Incs. In this work, to identify novel *C. trachomatis* Incs interfering with eukaryotic trafficking and/or targeting eukaryotic organelles, we ectopically expressed the genes encoding the predicted cytosolic regions of Inc proteins in *Saccharomyces cerevisiae*. By performing vacuolar protein sorting (Vps) assays in yeast and analysis of protein localization by fluorescence microscopy we identified two Incs causing Vps defects in yeast and seven Incs showing tropism for eukaryotic organelles. In particular, the first 88 amino acid residues within Inc CT006 co-localized with LDs and with the ER both in yeast and in mammalian HeLa cells. Cells infected by a *C. trachomatis* strain overproducing CT006 seemed to enhance the accumulation of host cell LDs within the region of chlamydial inclusions. However, this effect did not depend on the LD-targeting regions within CT006. As we found that the amino-terminal region of CT006 associates with LDs in transfected cells, a region which is exposed to the host cell cytosol during infection, our results suggest that CT006 could be involved in the interaction between the inclusion and host cell LDs.

## Results

### *C. trachomatis* Incs cause vacuolar protein sorting mistrafficking in yeast

To search for novel *C. trachomatis* Incs interfering with eukaryotic vesicular trafficking, we performed a functional screen using the yeast *S. cerevisiae* as a eukaryotic model. For this purpose, we focused on Incs which have been shown to localize at the inclusion membrane and with amino or carboxy-terminal regions longer than 40 amino acid residues predicted to be exposed to the host cell cytosol. We then generated *S. cerevisiae* NSY01 reporter strains (detailed in S1 Table) producing the predicted cytosolic domains of Incs fused to the green fluorescent protein (GFP) (Inc-GFP) (Fig 1a). It has been shown that for some Incs, the production of their cytosolic domains is sufficient to cause phenotypes in yeast, while other Inc fragments need to be fused to the localization signal (L) and transmembrane domains (TM) of the yeast SNARE Pep12p ($Pep12_{L-TM}$), which anchors the fragments to the cytosolic side of endosomes [8, 27] to mimic their presence in a membrane and thus be able to exert their function. Therefore, we also generated NSY01 reporter strains (S1 Table) producing the predicted cytosolic domains of Incs fused to GFP and to $Pep12_{L-TM}$ (Inc-GFP-$Pep12_{L-TM}$) (Fig 1a). In all strains, the genes encoding Inc-GFP and Inc-GFP-$Pep12_{L-TM}$ fusion proteins were expressed under the control of a galactose-inducible promoter. Immunoblotting revealed that most fusion proteins were produced and migrated on SDS-PAGE according to their predicted molecular mass (S1 and S2 Figs and summary of protein production in S2 and S3 Tables).

The generated yeast strains were then used to screen for Incs causing vacuolar protein sorting (Vps) defects in yeast, based on the ability of the *S. cerevisiae* NSY01 reporter strain to produce a modified form of invertase (Carboxipeptidase Y-Invertase; CPY-Inv). CPY-Inv normally traffics to the yeast vacuole, but can be secreted to the outside of the cell due to Vps mistrafficking (Fig 1b) [28]. As illustrated in Fig 1b, normal (white colonies; $Vps^+$ phenotype) or mistrafficking (brown colonies; $Vps^-$ phenotype) can be scored qualitatively [28].

A representative result of the Vps assays performed is illustrated in Fig 1c (all data are shown in S3 and S4 Figs and are summarized in S2 and S3 Tables). Yeast strains producing only GFP or GFP-$Pep12_{L-TM}$ were used as negative controls, and yeast strains producing proteins previously shown to induce a $Vps^-$ phenotype, the dominant-negative form of the yeast ATPase Vps4 ($Vps4^{E233Q}$) [28] and a fusion of the *Legionella pneumophila* effector VipA to GFP (VipA-GFP) [29], were used as positive controls. Using this approach, we found 2 fusion proteins (out of 61) causing a $Vps^-$ phenotype: $CT229_{91-215}$-GFP and $CT223_{192-268}$-GFP-$Pep12_{L-TM}$ (Fig 1c). CT229 (CpoS) and CT223 (IPAM) have been previously shown to interfere with host cell trafficking during *C. trachomatis* infection [18–22, 30]. Therefore, although we only identified CT223 and CT229 as capable of causing sorting defects in yeast, these results validate the applicability of the yeast Vps assay to screen for *C. trachomatis* Incs interfering with eukaryotic trafficking.

### *C. trachomatis* Incs co-localize with yeast organelles

To screen for *C. trachomatis* Incs targeting eukaryotic organelles, we analyzed the intracellular localization of Inc-GFP fusion proteins in yeast, on the assumption that their tropism for eukaryotic organelles could give insights about their functions. As depicted in S2 Table, we were able to detect 19 Inc-GFP fusion proteins (out of 31) by fluorescence microscopy. Among these 19 Inc-GFP proteins, 12 were found spread in the yeast cytosol similarly to GFP alone (S5 Fig) and 7 appeared in a specific intracellular localization, suggesting tropism for endosomal compartments ($CT229_{91-215}$-GFP), mitochondria ($CT179_{53-170}$-GFP, $CT324_{119-303}$-GFP, $CT618_{1-212}$-GFP, $CT018_{1-90}$-GFP and $CT383_{157-243}$-GFP) or LDs ($CT006_{1-88}$-GFP) (examples

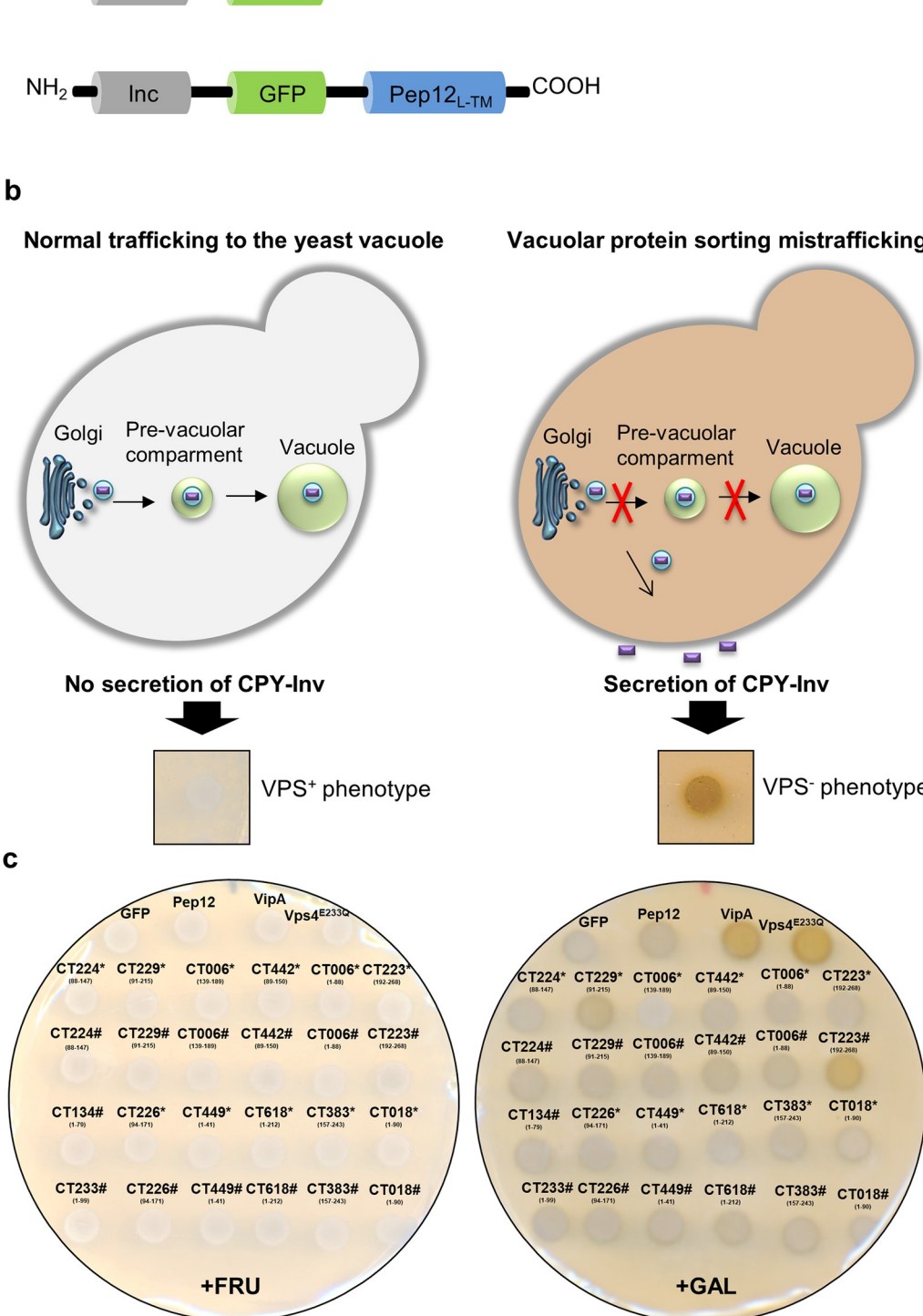

**Fig 1. *C. trachomatis* Incs cause vacuolar protein sorting mistrafficking in yeast.** (a) Schematic representation of the fusion proteins used in vacuolar protein sorting (Vps) assays in *Saccharomyces cerevisiae*. The *S. cerevisiae* NSY01 reporter strain (S1 Table) was transformed with plasmids encoding the predicted cytosolic domains of Incs fused to GFP (Inc-GFP) (left-hand side), or fused to GFP and also to the localization signal (L) and transmembrane domains (TM) of the yeast SNARE Pep12p (Inc-GFP-Pep12$_{L-TM}$) (right-hand side), under the control of a galactose-inducible promoter.

The list of *S. cerevisiae* NSY01 reporter strains producing *C. trachomatis* cytosolic Inc fragments fused to GFP or to GFP-Pep12$_{L-TM}$ is detailed in S1 Table. (b) The generated yeast strains were used to screen for *C. trachomatis* Incs causing Vps defects, detected by an assay based on the ability of the NSY01 reporter strain to produce caboxypetidase Y-Invertase (CPY-Inv), which hydrolyses sucrose into glucose and fructose at the cell surface when trafficking to the vacuole is disrupted. Normal (Vps$^+$ phenotype) or mistrafficking (Vps$^-$ phenotype) can be scored qualitatively in solid media using a sucrose overlay solution, indicating glucose production by formation of a brown precipitate (Vps$^-$ phenotype) [28]. (c) Representative results from the Vps assays in yeast. *S. cerevisiae* strains producing the indicated Inc fragments were grown in solid media under inducing (galactose; +GAL) or non-inducing (fructose; +FRU) conditions. After 48 h, the Vps phenotype was analyzed qualitatively in solid media. Two fusion proteins containing Inc fragments caused a Vps- phenotype: CT229$_{91-215}$-GFP and CT223$_{192-268}$-GFP-Pep12$_{L-TM}$. GFP and GFP-Pep12$_{L-TM}$ were used as negative controls (Vps$^+$ phenotype) and the dominant-negative form of the yeast ATPase Vps4 (Vps4$^{E233Q}$) [28] and a fusion to GFP of the *Legionella pneumophila* effector VipA (ViPA-GFP) [29] were used as positive controls (Vps$^-$ phenotype). * Represents Inc fragments fused to GFP and # represents Inc fragments fused to GFP-Pep12$_{L-TM}$. Vps results with all yeast strains analyzed are shown in S3 and S4 Figs and summarized in S2 and S3 Tables.

in Fig a in S5 Fig). Among the 5 Inc-GFP proteins that localized at mitochondria-like puncta, CT179$_{53-170}$-GFP and CT324$_{119-303}$-GFP migrated on SDS-PAGE according to their predicted molecular mass, while CT618$_{1-212}$-GFP, CT018$_{1-90}$-GFP and CT383$_{157-243}$-GFP migrated below their expected molecular mass (Fig a in S1 Fig and summary in S2 Table). As the majority of mitochondrial matrix proteins contain an amino-terminal targeting signal that is cleaved upon import [31], these differences might be due to differences in the length of the signal peptide that is cleaved.

To corroborate the observations on the localization of Incs in yeast, co-localization analyses by fluorescence microscopy between selected Inc-GFP proteins and organelle markers were performed. A yeast strain ectopically producing CT179$_{53-170}$-GFP was stained with MitoRed, a mitochondrial probe, which showed co-localization of CT179$_{53-170}$-GFP with mitochondria (Fig 2a). A yeast strain producing CT229$_{91-215}$-GFP, which induced a vacuolar protein traffic defect in yeast (Fig 1c), was incubated with FM4-64 (a fluorescent endocytic probe). This showed that CT229$_{91-215}$-GFP co-localizes with endosomal compartments (Fig 2b). GFP-Pep12$_{L-TM}$ and Inc-GFP-Pep12$_{L-TM}$ fusion proteins localized predominantly at endosomal compartments, such as endosome-like puncta and vacuoles (Fig b in S5 Fig and summary in S3 Table), as previously described [8, 27]. Finally, to test if CT006$_{1-88}$-GFP localized at LDs, we generated a yeast strain producing Erg6-mCherry [32] (protein marker of LDs) and CT006$_{1-88}$-GFP. This confirmed that CT006$_{1-88}$-GFP co-localizes with LDs (Fig 2c). We also analyzed yeast strains co-producing Erg6-mCherry and GFP or CT006$_{139-189}$-GFP, which revealed that CT006$_{139-189}$-GFP has a cytosolic localization as GFP alone (Fig 2c).

In summary, we found seven *C. trachomatis* Inc fragments associating with eukaryotic organelles. Five Incs associated with mitochondria but the relevance of this is unclear as a possible direct *C. trachomatis*-mitochondria interaction during infection has not been described. However, we found that CT006$_{1-88}$ localizes at LDs (Fig 2c), which have been shown to be targeted by several intracellular pathogens, including *C. trachomatis* (reviewed in [33]). Therefore, we selected CT006 for further characterization.

## The first 88 amino acid residues of CT006 fused to mEGFP co-localize with LDs in mammalian cells

The primary structure of CT006 (189 amino acid residues; Fig 3a) does not display significant similarities to known non-chlamydial proteins or domains. CT006 has a predicted bilobed hydrophobic motif, characteristic of Incs, between a tyrosine residue at position 89 (Y$_{89}$) and a histidine residue at position 140 (H$_{140}$) [16], which is composed of two transmembrane segments separated by a loop of 6 residues (Fig 3a and S6 Fig). CT006 is predicted to have an

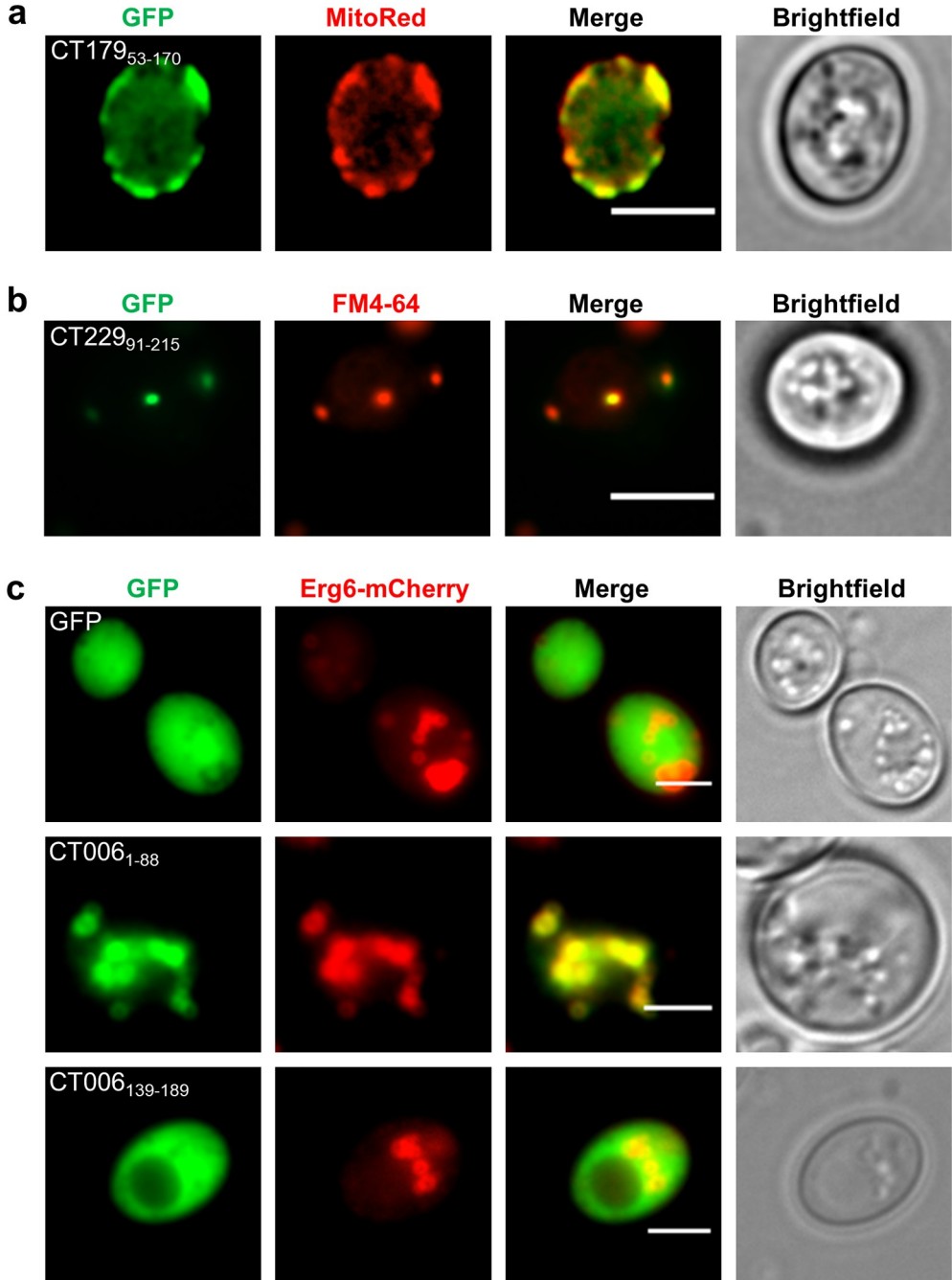

**Fig 2. Intracellular localization of *C. trachomatis* Inc fragments fused to GFP when produced in yeast.** *S. cerevisiae* strains ectopically expressing the genes encoding the indicated fusion proteins were grown in the presence of galactose. Live cells were visualized by fluorescence microscopy, directly or after the indicated staining. The intracellular localization of all Inc-GFP fusion proteins analyzed in this study is summarized in S2 Table. Scale bars, 5 μm. (a) Yeast strains producing CT179$_{53-170}$-GFP were stained with MitoRed, a mitochondrial probe, indicating that CT179$_{53-170}$-GFP co-localizes with mitochondria. (b) Yeast strains producing CT229$_{91-215}$ were incubated with FM4-64 and CT229$_{91-215}$ co-localized with FM4-64-stained endosomal compartments. (c) This analysis was performed with yeast strains ectopically co-expressing the genes encoding Erg6-mCherry (protein marker of LDs) and GFP, CT006$_{1-88}$-GFP or CT006$_{139-189}$-GFP and indicates that CT006$_{1-88}$-GFP co-localizes with LDs and CT006$_{139-189}$-GFP is cytosolic.

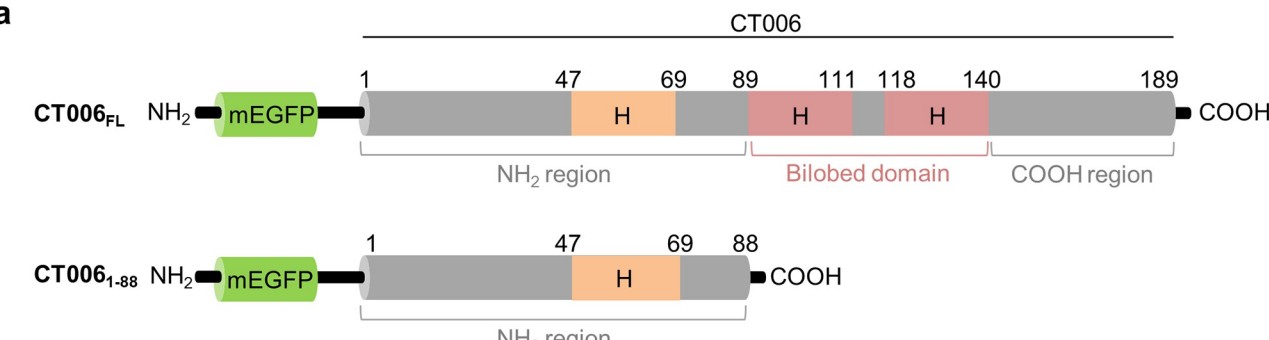

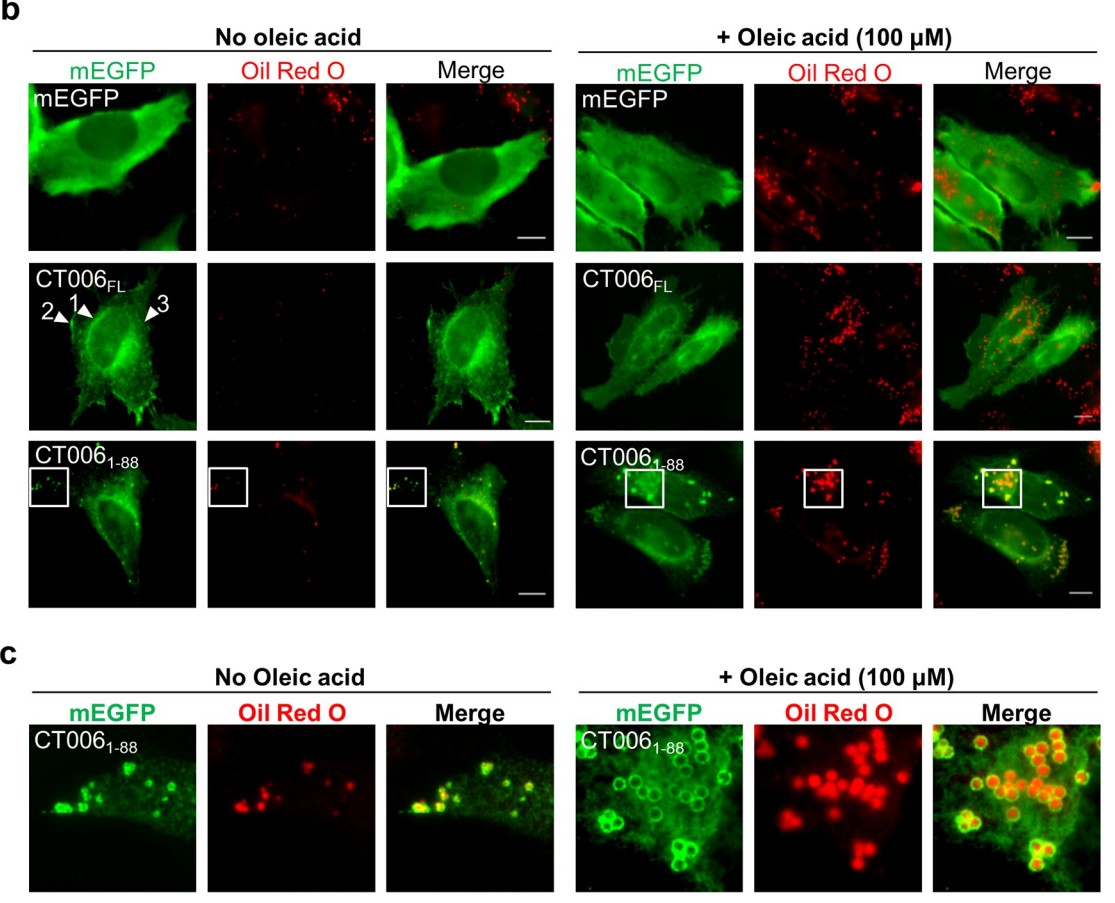

**Fig 3. The first 88 amino acid residues of CT006 fused to mEGFP co-localize with LDs in mammalian cells.** HeLa 229 cells were transfected with plasmids encoding mEGFP or different regions of CT006 containing a mEGFP tag at their amino-termini (mEGFP-CT006 proteins). (a) Schematic representation of mEGFP-CT006 and mEGFP-CT006$_{1-88}$ (not drawn to scale); H, Putative hydrophobic domain. (b) At 18 h post-transfection, cells were either treated with ethanol (solvent control; left-hand side panel) or 100 μM oleic acid (right-hand side panel) for 6 h and then fixed with 4% (w/v) PFA. Fixed cells were labeled with anti-GFP and the appropriate fluorophore-conjugated secondary antibody, stained with Oil Red O (3:2 v/v Oil Red O stock solution diluted in water), and imaged by fluorescence microscopy. Scale bars, 10 μm. Arrowheads indicate the reticular distribution of mEGFP-CT006$_{FL}$ (1) and the accumulation in patches near the plasma membrane (2) or cytosol (3). (c) In the area delimited by white squares (Fig 3b) images were zoomed.

additional hydrophobic and putative transmembrane domain between a glycine residue at position 47 ($G_{47}$) and a phenylalanine residue at position 69 ($F_{69}$) (Fig 3a and S6 Fig).

To test whether CT006 localizes at LDs in mammalian cells, HeLa 229 cells were transfected with plasmids encoding full length CT006 ($CT006_{FL}$), its first 88 amino acid residues ($CT006_{1-88}$), or its last 51 amino acid residues ($CT006_{139-189}$) fused to the amino- or carboxy-terminus of monomeric enhanced green fluorescent protein (mEGFP). Fluorescence microscopy revealed that the intracellular localization of CT006 proteins fused to mEGFP is independent of the position of the mEGFP tag (Fig a in S7 Fig). Immunoblotting of whole cell extracts from the transfected cells confirmed the production of fusion proteins with the predicted molecular mass but revealed less degradation products for mEGFP-CT006 versions comparing with CT006-mEGFP versions (Fig b in S7 Fig). Therefore, subsequent analyses were performed with mEGFP-CT006 versions. $mEGFP\text{-}CT006_{139-189}$ was not further analyzed, because it showed a cytosolic distribution as mEGFP alone (Fig a in S7 Fig).

As HeLa 229 cells are poor in LDs, in our experiments LDs synthesis was induced by the addition of 100 μM of oleic acid [5]. Briefly, HeLa cells ectopically expressing the genes encoding mEGFP, $mEGFP\text{-}CT006_{FL}$ or $mEGFP\text{-}CT006_{1-88}$ were either incubated with the ethanol solvent alone (Fig 3b; left-hand side panel) or with 100 μM oleic acid (Fig 3b; right-hand side panel) for 6 h before fixation. The LDs were stained with the neutral lipid dye Oil Red O and the cells were analyzed by fluorescence microscopy. As expected, the incubation with oleic acid increased the number and size of LDs in HeLa cells, comparing with cells incubated only with the solvent (Fig 3b). The $mEGFP\text{-}CT006_{FL}$ protein did not co-localize with LDs, but instead it showed inconsistent localizations within cells, including a reticular distribution and an accumulation in puncta and patches in the cytosol and also near the plasma membrane (Fig 3b). Moreover, the localization of $mEGFP\text{-}CT006_{FL}$ remained unaltered regardless of the incubation with oleic acid. In contrast, the $mEGFP\text{-}CT006_{1-88}$ protein besides retaining some reticular distribution, partially appeared as circles surrounding Oil Red O-stained LDs (Fig 3b). Incubation with oleic acid led to an increase in number and size of LDs in all cells examined (Fig 3b), and, as shown in the zoomed images in Fig 3c, changed the localization of $mEGFP\text{-}CT006_{1-88}$ from small to large circles, all surrounding Oil Red O-stained LDs (Fig 3c).

To clarify the reticular distribution of CT006, HeLa cells producing $mEGFP\text{-}CT006_{FL}$ or $mEGFP\text{-}CT006_{1-88}$ were immunolabeled with an antibody against protein disulfide isomerase (PDI), revealing that both proteins partially co-localize with the ER (S8 Fig).

## Positively charged amino acids within $CT006_{1-88}$ are essential to target $mEGFP\text{-}CT006_{1-88}$ to LDs in mammalian cells

It was previously shown that association of caveolin with LDs is mediated by two motifs acting cooperatively [34]. A central hydrophobic domain anchors caveolin to the ER and then positively-charged amino acid residues mediate its sorting to LDs [34]. Therefore, we searched for positively charged residues near the putative hydrophobic domain in $CT006_{1-88}$ (Fig 3a and S6 Fig) and five candidates were identified: two lysine residues ($K_{34}$, $K_{37}$) upstream from the hydrophobic domain and arginine, histidine, and lysine ($R_{72}$, $H_{80}$, $K_{81}$) residues localized downstream from the hydrophobic domain (Fig 4a). To test for a role of these residues in targeting $CT006_{1-88}$ to LDs, we generated mammalian transfection plasmids encoding $mEGFP\text{-}CT006_{1-88}$ versions where the identified residues were replaced in different combinations by neutral glycine residues (K34G, K37G; H80G, K81G; R72G, H80G, K81G; K34G, K37G, H80G, K81G; or K34G, K37G, R72G, H80G, K81G). After transfection of HeLa cells with these plasmids we confirmed by immunoblotting that the mutant proteins were produced and migrated on SDS-PAGE according to their predicted molecular mass (S9 Fig). HeLa cells

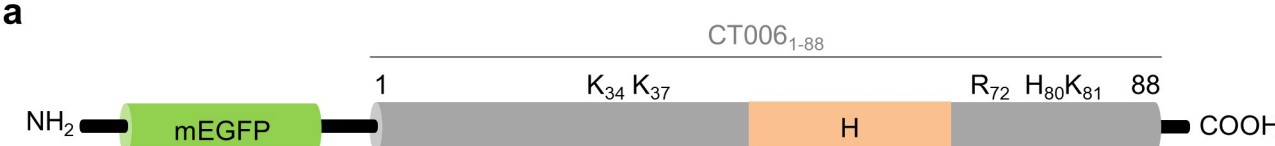

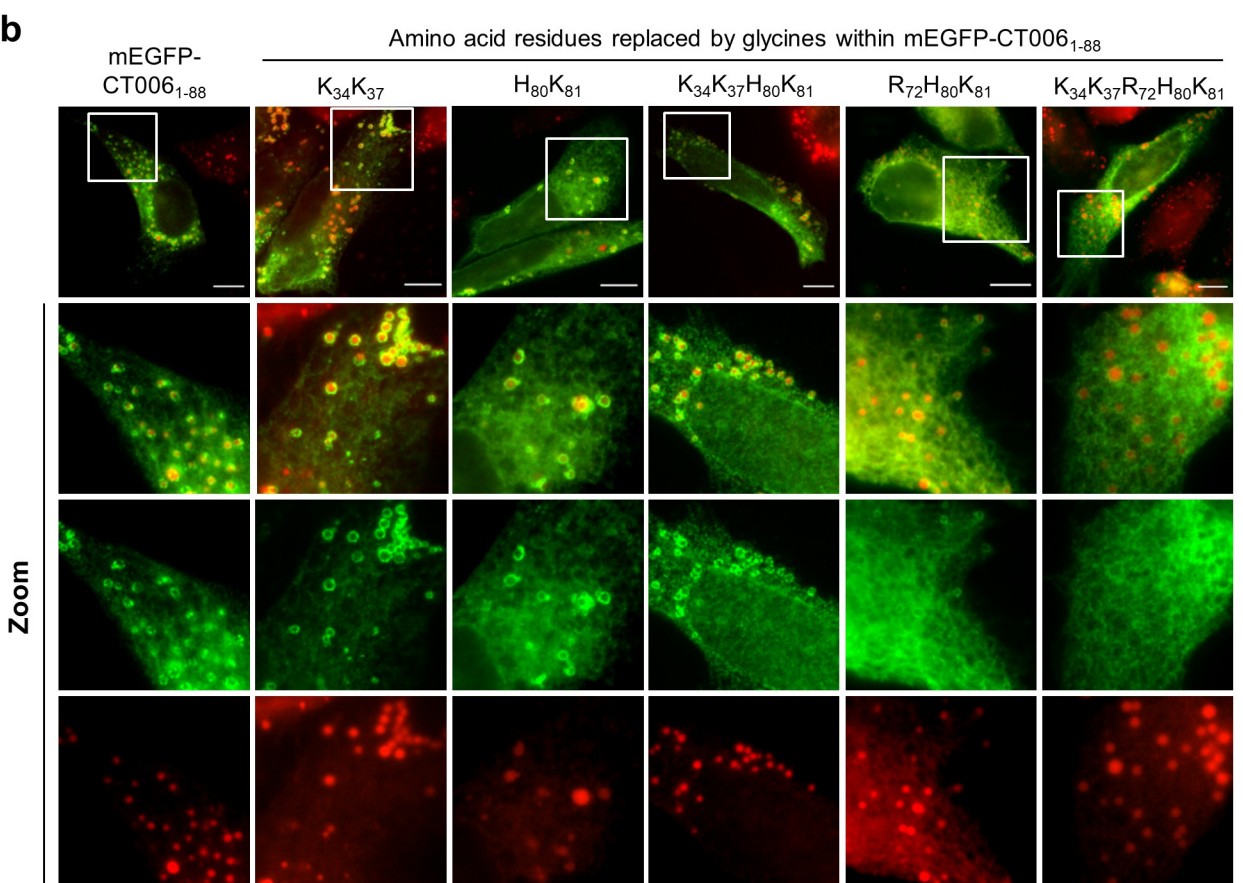

**Fig 4. Positively charged amino acids within CT006$_{1-88}$ are important to target mEGFP-CT006$_{1-88}$ to LDs in mammalian cells.** (a) Schematic representation of the position of positively charged amino acids within CT006$_{1-88}$ (not drawn to scale); H, Putative hydrophobic domain. (b) HeLa 229 cells were transfected with plasmids encoding mEGFP or different mEGFP-CT006$_{1-88}$ versions where the indicated amino acid residues were replaced by glycines (G). At 18 h post-transfection, cells were treated with 100 µM oleic acid for 6 h and fixed with 4% (w/v) PFA. Fixed cells were labeled with anti-GFP and the appropriate fluorophore-conjugated secondary antibody, stained with Oil Red O (3:2 v/v Oil Red O stock solution diluted in water), and imaged by fluorescence microscopy. Scale bars, 10 µm. In the areas delimited by white squares images were zoomed.

ectopically expressing the genes encoding mEGFP, mEGFP-CT006$_{1-88}$ or mEGFP-CT006$_{1-88}$ variants with amino acid replacements were incubated with oleic acid to induce LD synthesis. The cells were then fixed, stained with Oil Red O and analyzed by fluorescence microscopy. The substitutions for glycines of K$_{34}$K$_{37}$, H$_{80}$K$_{81}$ or K$_{34}$K$_{37}$H$_{80}$K$_{81}$ resulted in mutant proteins localizing at LDs as mEGFP-CT006$_{1-88}$ (Fig 4b). When the three positively charged amino acid residues (R$_{72}$H$_{80}$K$_{81}$) downstream from the hydrophobic domain were replaced by glycines, the protein became predominantly reticulated, but still retained some tropism for LDs. In contrast, when the five positively charged amino acid residues (K$_{34}$K$_{37}$R$_{72}$H$_{80}$K$_{81}$) were simultaneously replaced by glycines, the mutant protein showed almost solely a reticular distribution

without defined circles surrounding LDs (Fig 4b), suggesting that its sorting to LDs was substantially reduced. This analysis revealed that positively charged sequences close to the amino-terminal hydrophobic domain of CT006 (Figs 3a and 4a) are important to target mEGFP-CT006$_{1\text{-}88}$ to LDs.

## The putative cytosolic regions of CT006 are exposed to the host cell cytosol

As the first 88 amino acids of CT006 can mediate an association with LDs when this fragment is ectopically produced in eukaryotic cells, we sought to understand if this region of the protein is exposed to the host cytosol in infected cells. For this, we used *C. trachomatis* strains producing CT006 or control proteins tagged with a 13-residue phosphorylatable peptide from glycogen synthase kinase (GSK)-3β [35]. This peptide is derived from GSK-3β and when fused to bacterial proteins has been used to identify chlamydial and other bacterial effectors and to confirm the cytosolic exposure of Incs [35–37]. The GSK tag is phosphorylated by cytosolic eukaryotic protein kinases, allowing to use immunoblotting with phospho-specific GSK antibodies as indicative of protein localization outside of the chlamydial inclusion. Therefore, we transformed *C. trachomatis* serovar L2 strain 434/Bu (L2/434) with plasmids encoding different versions of CT006 tagged with GSK under the control of the *ct006* promoter. Specifically, to test if the carboxy-terminal region of CT006 was exposed to the cell cytosol, GSK was fused to CT006 after the last amino acid residue [CT006-GSK(189)]. As tagging GSK to the amino-terminus of CT006 should interfere with the T3S signal, to test the localization of the amino-terminal region we generated *C. trachomatis* strains harboring fusion proteins with GSK integrated upstream from the putative hydrophobic domain between amino acid residues 47 and 69 (Fig 3a and S6 Fig), specifically after the first 26 [CT006-GSK(26)] or 39 [CT006-GSK(39)] amino acid residues of CT006. *C. trachomatis* L2/434-derived strains producing the *C. trachomatis* ribosomal protein RplJ/CT317 or the *C. trachomatis* T3S effector CteG [38] with a carboxy-terminal GSK tag (RplJ-GSK or CteG-GSK) were generated and used as negative or positive control, respectively. The genes encoding RplJ-GSK or CteG-GSK were expressed under the control of the tetracycline inducible *tet* promoter.

HeLa 229 cells were then infected with the different *C. trachomatis* L2/434-derived strains encoding GSK-tagged chlamydial proteins. After 24 h, infected cells were harvested and analyzed by immunoblotting using antibodies against total GSK (anti-GSK3β) or against phosphorylated GSK (anti-pGSK3β). Detection with the anti-GSK3β antibody revealed production of RplJ-GSK, CteG-GSK, CT006-GSK(189), and CT006-GSK(26) (Fig 5a). In contrast, CT006-GSK(39) could not be detected (Fig 5a) indicating that this tagged protein is likely unstable. Immunoblotting using anti-pGSK3β revealed the expected phosphorylation of CteG-GSK (Fig 5a), indicative of localization in the host cell cytosol, and lack of phosphorylation of RplJ-GSK (Fig 5a), confirming its retention within the inclusion, presumably inside bacteria. The results with these controls confirmed the validity of the assay. As CT006-GSK (189) and CT006-GSK(26) were both detected by the anti-pGSK3β antibody (Fig 5a), both the amino and the carboxy termini of CT006 are exposed to the host cell cytoplasm, outside of the inclusion. By indirect immunofluorescence microscopy using anti-pGSK3β antibodies we further showed that both CT006-GSK(189) and CT006-GSK(26) concentrate at the inclusion membrane, while only background signal was detected for cells infected by L2/434 or by L2/434 producing RplJ-GSK (Fig 5b).

Altogether, these results indicate that the central bilobed hydrophobic domain of CT006 mediates the insertion of CT006 at the inclusion membrane, whereas its amino and carboxy-terminal regions are exposed to the host cell cytosol (Fig 5c). Therefore, the putative hydrophobic domain of CT006 between amino acid residues 47 and 69 (Fig 3a and S6 Fig), should

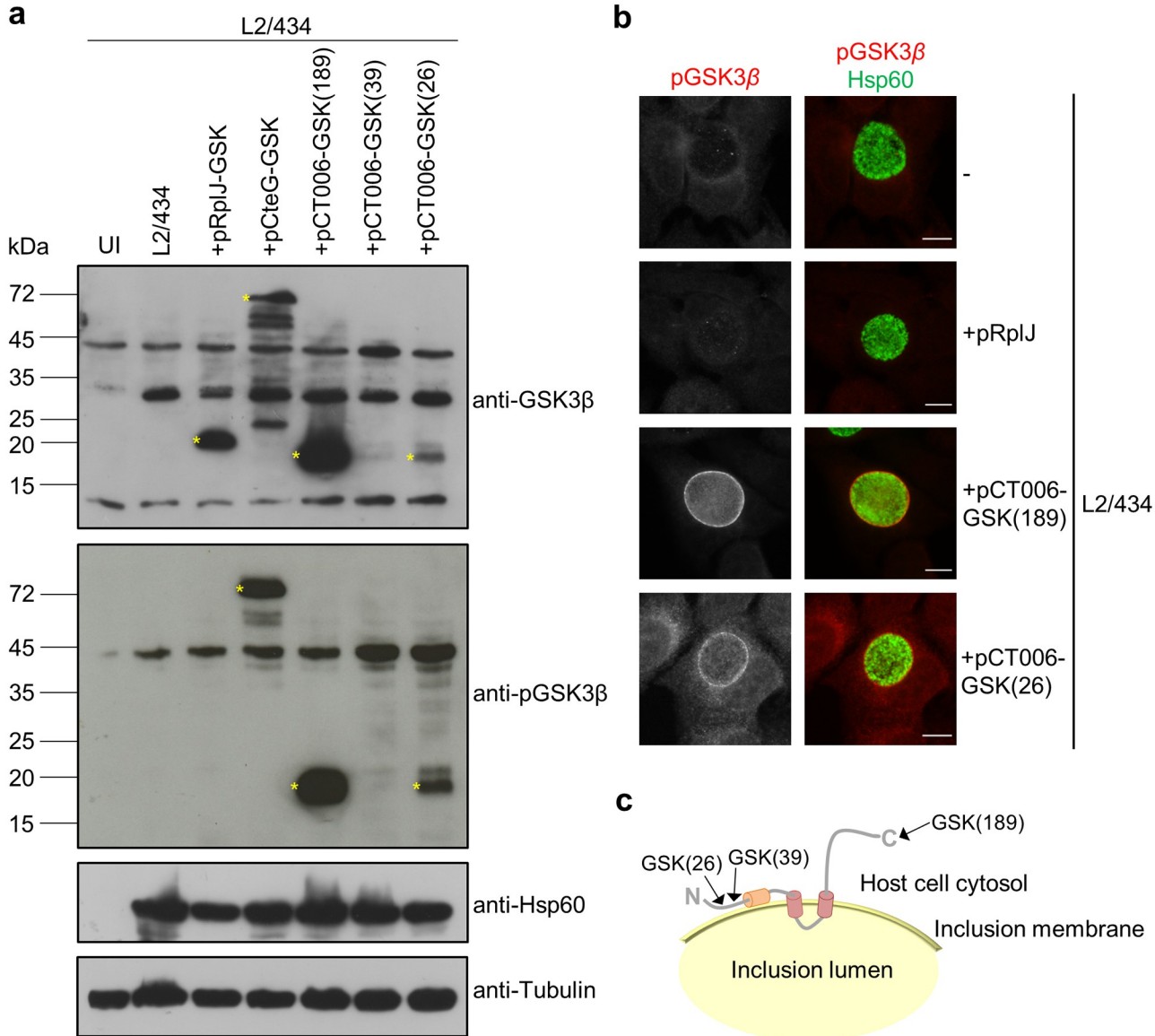

**Fig 5. Analysis of the topology of CT006 at the inclusion membrane.** HeLa 229 cells were left uninfected (UI) or infected by *C. trachomatis* strains L2/434, L2/434+pRplJ-GSK, L2/434+pCteG-GSK, L2/434+pCT006-GSK(26), L2/434+pCT006-GSK(39) or L2/434+pCT006-GSK(189). (a) At 24 h post-infection, whole cell extracts were analyzed by immunoblotting using the antibodies anti-GSK3β, anti-phospho GSK3β (anti-pGSK3β), anti-*C. trachomatis* Hsp60 (bacterial loading control) and anti-α-tubulin (HeLa 229 cells loading control) and the appropriate horseradish peroxidase (HRP)-conjugated secondary antibodies, followed by detection using SuperSignal West Femto (GSK and pGSK) or SuperSignal West Pico detection kit (Hsp60 and tubulin) (Thermo Fisher Scientific). (b) At 24 h post-infection, cells were fixed with 4% (w/v) PFA, immunolabeled with anti-pGSK3β (red) and anti-Hsp60 (green), and appropriate fluorophore-conjugated secondary antibodies. The labeled cells were then imaged by fluorescence microscopy. Scale bars, 10 μm. (c) Schematic representation of the deduced topology of CT006 at the inclusion membrane. The arrows indicate the position where the GSK tag was fused. The analysis with the GSK tag showed that the putative hydrophobic domain of CT006 between amino acid residues 47 and 69 (in orange) does not cross the inclusion membrane.

not cross the inclusion membrane (Fig 5c). Given its hydrophobicity, we speculate that this domain could insert in the leaflet of the inclusion membrane facing the host cell cytoplasm leaving one hydrophobic surface available to interact with LDs or with regions of the ER from where LDs originate.

## Characterization of CT006-2HA during *C. trachomatis* infection

To understand the role of CT006 during infection of tissue culture cells by *C. trachomatis*, we attempted to generate a *C. trachomatis ct006* null mutant by group II intron-based insertional mutagenesis [39] or by fluorescence-reported allelic exchange mutagenesis (FRAEM) [40]. However, after several attempts none of these approaches was successful. To study CT006 during *C. trachomatis* infection, we generated a *C. trachomatis* L2/434 strain harboring a plasmid encoding CT006 with a double hemagglutinin (2HA) tag at its carboxy-terminus (pCT006-2HA) (L2/434+pCT006-2HA strain), under the control of the *ct006* promoter. This strain produced CT006 from the chromosome and CT006-2HA from an expression vector derived from the endogenous *C. trachomatis* virulence plasmid. Previous studies with other *C. trachomatis* T3S substrate genes (*ct142*, *ct143*, and *cteG*) expressed from the same backbone plasmid (pSVP247) [38, 41] reported a ~10-fold increase in total mRNA and protein levels relative to the endogenous genes and proteins. The levels of CT006-2HA in the L2/434+pCT006-2HA strain are thus expected to be much higher than of CT006 in the L2/434 parental strain.

The *C. trachomatis* strain producing CT006-2HA was then characterized. Immunoblotting of extracts of infected HeLa cells revealed that CT006-2HA was detected from 16 to 44 hours post-infection and migrated on SDS-PAGE according to its predicted molecular mass (23 kDa) (Fig 6a). Indirect immunofluorescence microscopy of HeLa cells infected by L2/434 +pCT006-2HA confirmed that CT006-2HA concentrates at the inclusion membrane and co-localizes with *C. trachomatis* Cap1 (known to localize at the inclusion membrane [42]) (Fig 6b). Additional experiments where L2/434+pCT006-2HA was used to infect HeLa cells followed by analysis by indirect immunofluorescence microscopy, revealed that CT006-2HA was produced from 2 h post-infection and accumulated around the inclusion from at least 16 h post-infection (S10 Fig). Furthermore, delivery of CT006-2HA into host cells could be detected from 2 h post-infection, while accumulation around the early inclusion was not always obvious (S10 Fig). At 8 h post-infection CT006-2HA appeared in small tubules near the inclusion, suggesting that these tubules could be an extension of the vacuolar membrane (S10 Fig). Overall, this confirmed that CT006 is a bona fide Inc, as previously shown [17], which is delivered into host cells at least from 2 h post-infection and accumulates at the inclusion membrane until late in the developmental cycle. Finally, to investigate whether plasmid-encoded CT006-2HA could affect the intracellular growth of *C. trachomatis* in HeLa cells, the generation of infectious progeny and the area of the inclusions was compared between the parental and the L2/434+pCT006-2HA strains at 44 and 24 h post-infection, respectively. However, no significant differences were observed (Fig 6c and 6d), indicating that overproduction of CT006-2HA does not inhibit or promote *C. trachomatis* growth.

## The effect of CT006-2HA produced by *C. trachomatis* on host cell LDs

LDs are observed in close association with the inclusion membrane in HeLa cells infected by *C. trachomatis* L2/434 [5]. We questioned if increased production of plasmid-encoded CT006-2HA by *C. trachomatis* relative to the parental L2/434 strain could facilitate the association of LDs with the inclusion. To address this, we sought to use fluorescence microscopy images of cells infected by *C. trachomatis* for 10, 14, 18 and 22 h and quantify the area of LDs at the region of the inclusion. We reasoned that a possible effect of CT006-2HA overproduction in the association of LDs with the inclusion should not be seen by mutant CT006 proteins with defects in the possible LD-targeting region and neither by a random Inc. Therefore, we generated three additional L2/434-derived strains: L2/434+pCT006$_{5G}$-2HA, harboring a plasmid-encoded CT006$_{5G}$-2HA mutant protein where the amino acid residues important for the localization of mEGFP-CT006$_{1-88}$ at LDs in transfected cells ($K_{34}K_{37}R_{72}H_{80}K_{81}$) were replaced by

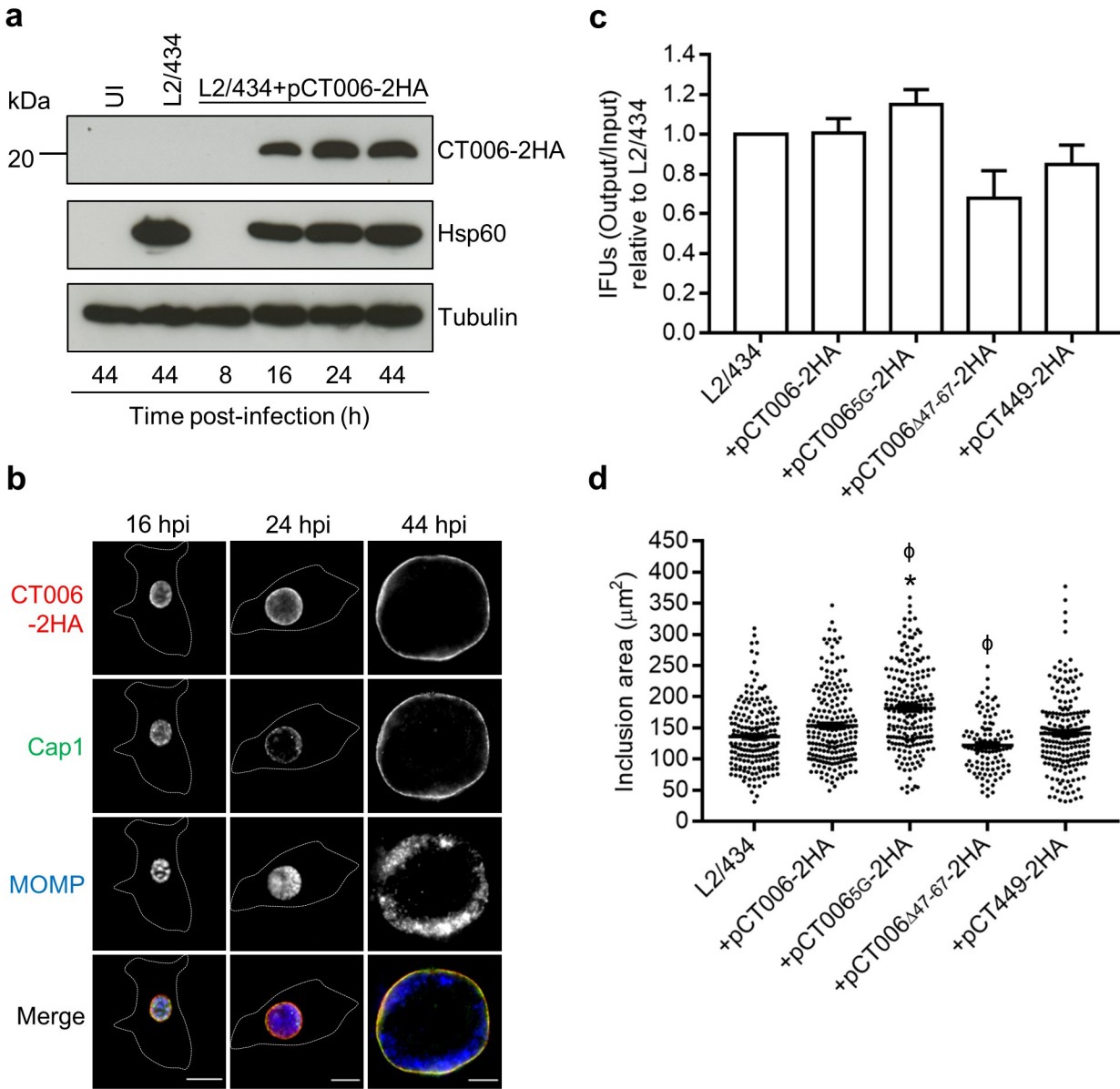

**Fig 6. Characterization of *C. trachomatis* L2/434 strain harboring pCT006-2HA.** HeLa 229 cells were left uninfected (UI) or infected by the *C. trachomatis* L2/434 parental strain or by *C. trachomatis* L2/434 strains harboring pCT006-2HA (L2/434+pCT006-2HA), pCT006$_{5G}$-2HA (L2/434 +pCT006$_{5G}$-2HA), pCT006$_{\Delta 47-67}$-2HA (L2/434+pCT006$_{\Delta 47-67}$-2HA) or pCT449-2HA (L2/434+pCT449-2HA). (a) At the indicated hours post-infection (hpi), whole cell extracts were analyzed by immunoblotting using antibodies against HA, Hsp60 (bacterial loading control) and α-tubulin (HeLa 229 cells loading control) and the appropriate HRP-conjugated secondary antibodies, followed by detection using SuperSignal West Pico detection kit (Thermo Fisher Scientific). (b) At the indicated hours post-infection (hpi), infected cells were fixed with 4% (w/v) PFA, immunolabeled with antibodies against HA (red), *C. trachomatis* Cap1 (green), known to localize at the inclusion membrane [42], and *C. trachomatis* major outer membrane protein (MOMP) (blue), and appropriate fluorophore-conjugated secondary antibodies. The labeled cells were then imaged by fluorescence microscopy. Scale bars, 10 μm. Dashed lines represent the limits of infected HeLa cells. (c) Output inclusions forming units (IFUs) per input IFUs were calculated for each *C. trachomatis* strain at 44 h post-infection and divided by the values obtained for the parental L2/434 strain. Data are represented as the mean and standard error of the mean of at least 3 independent experiments. Statistical analysis was performed using Wilcoxon signed-rank test. (d) At 24 hpi, the area of chlamydial inclusions was measured for more than 100 particles randomly selected from images from at least 3 independent experiments using Fiji software [43]. Statistical analysis was performed using Kruskal-Wallis and Dunn's multiple comparisons test. * Represents P<0.05 by comparison to the L2/434 strain; φ represents P<0.05 by comparison to the L2/434 +pCT006-2HA strain.

glycines (Fig 4b); L2/434+pCT006$_{\Delta 47-67}$-2HA, harboring a plasmid-encoded CT006$_{\Delta 47-67}$-2HA mutant protein lacking amino acid residues 47 to 67, which comprise most of the putative hydrophobic motif possibly required for the association of CT006 with LDs (Fig 4a); and L2/434+pCT449-2HA, a strain harboring plasmid-encoded Inc CT449 with a carboxy-terminal 2HA tag. In these strains, the genes encoding the 2HA-tagged proteins are expressed from the corresponding *ct006* or *ct449* promoters.

We first confirmed that Inc CT449 does not show tropism for LDs in transfected HeLa cells (S11 Fig). Furthermore, during infection of HeLa cells by *C. trachomatis*, plasmid-encoded CT006$_{5G}$-2HA, CT006$_{\Delta 47-67}$-2HA, and CT449-2HA are produced and accumulate at the inclusion membrane (S12 and S13 Figs). The production of CT006$_{5G}$-2HA, CT006$_{\Delta 47-67}$-2HA, and CT449-2HA did not significantly affect the ability of *C. trachomatis* to generate infectious progeny (Fig 6c), but cells infected by the L2/434+pCT006$_{5G}$-2HA strain showed significantly larger inclusions than those infected by the L2/434 strain (Fig 6d). A comparison against the inclusions of cells infected by L2/434+pCT006-2HA also revealed that the production of CT006$_{5G}$-2HA resulted in larger inclusions, while the production of CT006$_{\Delta 47-67}$-2HA led to smaller inclusions (Fig 6d). As these strains were not significantly affected in their ability to generate infectious progeny, the differences in inclusion area might be related with alterations in inclusion morphology, and/or with the distribution of the chlamydiae within the inclusion, due to the accumulation of CT006 mutant proteins at the inclusion membrane.

Then, to test for a possible effect of the increased production of plasmid-encoded CT006-2HA on the distribution of LDs, HeLa cells were infected by *C. trachomatis* strains L2/434, L2/434+pCT006-2HA, L2/434+pCT006$_{5G}$-2HA, L2/434+pCT006$_{\Delta 47-67}$-2HA, or L2/434+pCT449-2HA. The cells were treated with oleic acid for 6 h, before fixation at 10, 14, 18 and 22 h post-infection. Fixed infected cells were labeled with anti-*C. trachomatis* Hsp60 and with the neutral lipid dye Bodipy 493/503 (which stains LDs) and analyzed by indirect immunofluorescence microscopy. At a first glance, the distribution of LDs was similar between cells infected by all *C. trachomatis* strains (Fig 7a and S14 Fig). To analyze this in further detail, the area of BODIPY-positive LDs within the region of chlamydial inclusions was measured from randomly selected images, as those depicted in Fig 7a and S14 Fig. For this analysis, both LDs localizing at the periphery of the inclusion or co-localizing with the inclusion were considered (see Materials and methods). We observed a minor, but significant, increase in the area of LDs within the inclusion region of cells infected by L2/434+pCT006-2HA comparing with cells infected by L2/434, or *vice-versa*, at 18 h (Fig 7b). There were no significant differences in the area of LDs within the inclusion region when comparing cells infected by strains producing CT006$_{\Delta 47-67}$-2HA, CT006$_{5G}$-2HA and CT449-2HA with cells infected by the L2/434 or the L2/434+pCT006-2HA strains (Fig 7b). Similar results were obtained in initial experiments with only some of the strains used in Fig 7b (S15 Fig). Therefore, while this data suggests that CT006 might participate in mediating the interaction of the inclusion with LDs, as only in cells infected by L2/434+pCT006-2HA significant differences were detected, it is not conclusive because the effect of CT006 overproduction in the area of LDs within the inclusion region did not show a dependency on the amino acid residues that promote the association of CT006 with LDs in transfected cells.

## Discussion

In this study, we found that *C. trachomatis* Inc CT006 associates with host cell LDs when its first 88 amino acid residues are ectopically produced in eukaryotic cells. Furthermore, we identified positively charged residues flanking a putative hydrophobic region that mediate the targeting of CT006$_{1-88}$ to LDs. We also found that this putative hydrophobic region should be

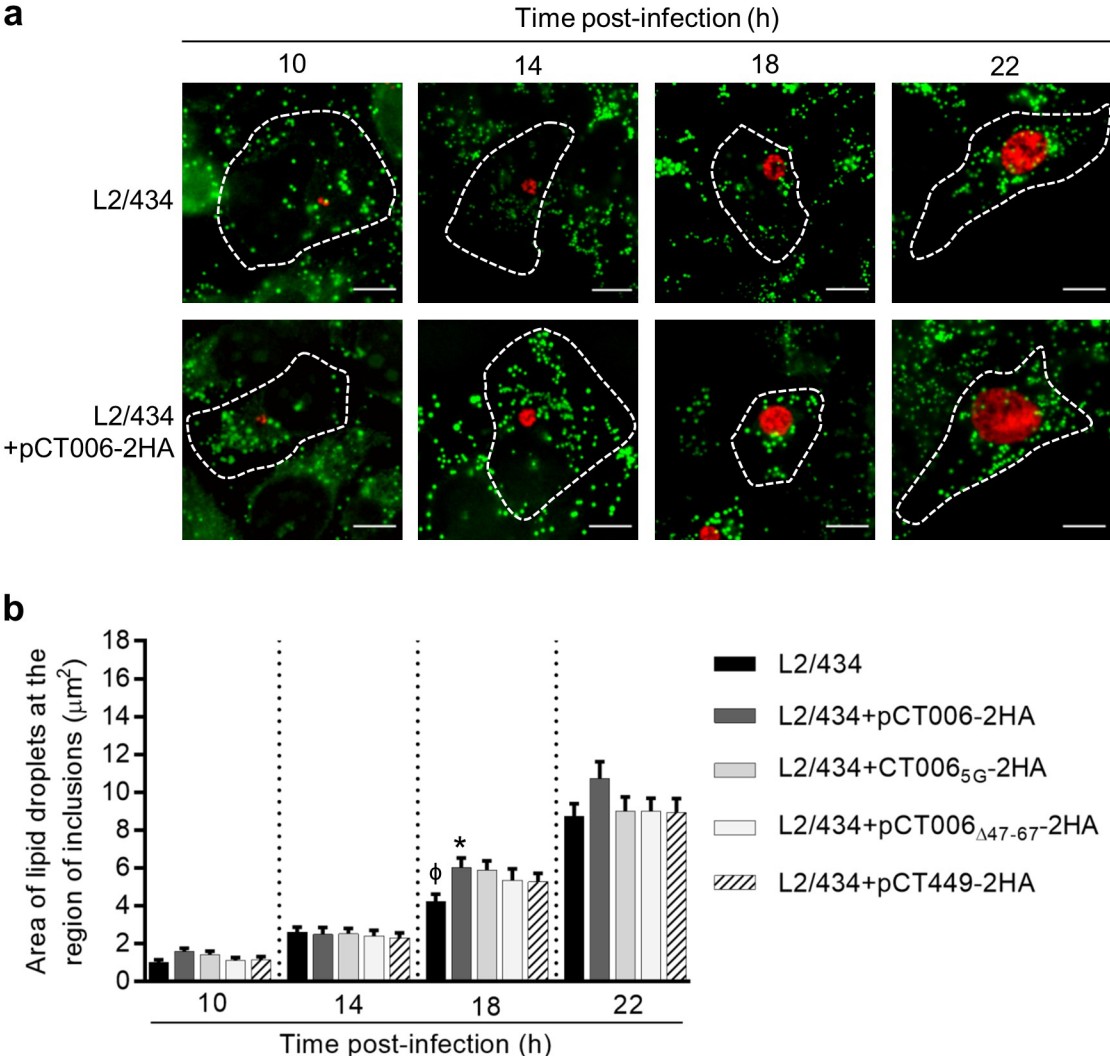

**Fig 7. The effect of CT006-2HA produced by *C. trachomatis* on host cell lipid droplets.** HeLa 229 cells were infected by the *C. trachomatis* L2/434 parental strain or by *C. trachomatis* L2/434 strains harboring pCT006-2HA (L2/434+pCT006-2HA), pCT006$_{5G}$-2HA (L2/434+pCT006$_{5G}$-2HA), pCT006$_{\Delta47\text{-}67}$-2HA (L2/434+pCT006$_{\Delta47\text{-}67}$-2HA) or pCT449-2HA (L2/434+pCT449-2HA). (a) At the indicated times post-infection, cells previously treated with 100 μM oleic acid for 6 h were fixed with 4% (w/v) PFA, immunolabeled with an antibody against Hsp60 (red) and an appropriate fluorophore-conjugated secondary antibody, stained with the neutral lipid dye BODIPY (green) and imaged by fluorescence microscopy. Scale bars, 10 μm. (b) The area of BODIPY-positive LDs at the region of chlamydial inclusions was measured from randomly selected images, as those depicted in Fig 7a and S14 Fig, using Fiji software [43]. Data are represented as the mean and standard error of the mean from three independent experiments (n = 60). Statistical analysis was performed using Kruskal-Wallis test with Dunn's multiple comparison test. * Represents P<0.05 by comparison to the L2/434 strain; ϕ represents P<0.05 by comparison to L2/434+pCT006-2HA.

exposed to the outside of the inclusion. Using a *C. trachomatis* strain producing plasmid-encoded CT006-2HA, we found that CT006-2HA accumulates at the inclusion membrane and slightly, but significantly, increased the area of LDs within the region of chlamydial inclusions relative to cells infected by the parental strain producing only chromosomal-encoded CT006. While this suggests that CT006 might participate in mediating the association of LDs with the *C. trachomatis* inclusion, the data was not conclusive as the area of LDs within the region of chlamydial inclusions of cells infected by *C. trachomatis* producing mutant CT006 with a defective LD-targeting region (or producing Inc CT449) was not different relative to cells

infected by the CT006-2HA overproducing strain. This must be clarified in future experiments when eventually a *C. trachomatis* strain with knocked-out, or transiently knocked-down, *ct006* becomes available.

This work started with a screen using *S. cerevisiae* as a model aiming to find novel functions for *C. trachomatis* Incs. With this approach, we identified fragments of two Incs (CT229/CpoS and CT223/IPAM) causing mistrafficking in yeast. As these Incs are known to modulate host cell trafficking related processes [19–22], this confirmed the applicability of the yeast Vps assay to screen for *C. trachomatis* Incs interfering with eukaryotic trafficking. Fine-tuning of the Inc fragments used in the assay must however be performed, as we could not detect production of many of the designed fusion proteins in yeast. Because we were looking for novel functions for Incs, we did not proceed with the analysis of CT229/CpoS and CT223/IPAM. Also, we do not exclude the possibility that other Incs are able to interfere with host cell vesicular trafficking as full-length proteins and in the context of infection. For instance, the fragment of CT119/IncA used was unable to cause Vps defects in yeast, but CT119/IncA inhibits endocytic membrane fusion during *C. trachomatis* infection [26]. In the screen in yeast, we also analyzed tropism for eukaryotic organelles and identified seven Incs whose putative cytosolic regions revealed relevant localizations, which was the starting point for the subsequent characterization of CT006. Overall, this study further emphasizes that a preliminary screen in yeast can be a valuable tool to expand the knowledge on chlamydial effectors [8, 44].

CT006 has a predicted hydrophobic domain within the LD-targeting region, and positively charged amino acids close to this domain are essential to target ectopically expressed $CT006_{1-88}$ to LDs. These eukaryotic organelles originate from the ER and, typically, eukaryotic class I proteins targeting LDs associate with membranes through hydrophobic hairpins and can localize both in the ER and LDs monolayer [45]. In the case of caveolin, the central hydrophobic domain anchors caveolin to the ER and then positively charged sequences mediate the sorting of caveolin to LDs [34], which is in agreement with our findings for $CT006_{1-88}$. In contrast, full-length CT006 showed a reticular distribution in HeLa cells, and also accumulated in puncta and patches in the cytosol and near the plasma membrane. Possibly, the bilobed hydrophobic domain of CT006 leads to protein aggregation or to incorrect targeting to cellular membranes. Somewhat in line with this, while previous studies using full-length Incs ectopically expressed in mammalian cells were useful to deduce or raise hypotheses about the functions of Incs, they also highlighted that Incs obviously exhibit a different localization than in infected cells [46, 47].

We speculate that in infected cells the same region and motifs that target $CT006_{1-88}$ to LDs might insert in the membrane leaflet of the inclusion membrane facing the host cell cytosol, similarly to monotopic membrane proteins, leaving one surface free to potentially interact with LDs or with regions of the ER originating LDs. While this needs to be tested, CT006 has a proline residue near the center of the hydrophobic domain in its amino-terminal region, possibly aiding in hairpin formation, which is also present in the metyltransferase AAM-B [48]. The amino-terminal region of AAM-B associates with LDs with both ends facing the cytosol, indicating the formation of a hairpin loop in the phospholipid monolayer surrounding LDs [48]. In addition to AAM-B, the proline residue was also detected within the hydrophobic domain of other LDs proteins (ALDI, CYB5R3), but its substitution by leucine did not affect targeting [48]. The carboxy-terminal of CT006 can potentially interact with other host cell organelles and proteins. Previous studies predicted the interaction of CT006 with eukaryotic 14-3-3 proteins [25], protein tyrosine kinase 7 (PTK7) [25] and the SNAREs Vamp3 and Vamp4 [49], but these interactions remain to be validated.

The *C. trachomatis* inclusion establishes direct contact with the ER. This is mediated by at least one Inc protein, IncV, which interacts with the ER-resident proteins VAPA and VAPB,

thus leading to the formation of ER-inclusion membrane contact sites [50]. Interestingly, the genes encoding CT006 and IncV are organized next to each other on the same strand and separated by 127 nucleotides in the *C. trachomatis* genomes. As the neighboring genes do not encode Incs, *ct006* and *incV* correspond to a small island of *inc* genes, supporting that the functions of the encoded proteins could be related. The generation of a *ct006* mutant strain (conditional if the gene is essential) is required to test if the encoded protein mediates recruitment of LDs to the inclusion and a possible functional relation with IncV. Unfortunately, we have been unable to generate such *ct006* mutant strain. This might be related with technical specificities of our experimental conditions or that *ct006* is an essential gene. The latter is supported by the fact that a previous large mutagenesis screen did not isolate nonsense mutations for *ct006* [24]. The recent development of a CRISPR interference system enabling induced and transient repression of chlamydial genes [51] should help clarifying this issue but this is beyond the scope of this manuscript.

Previously, other chlamydial proteins have been shown to associate with LDs. In particular, Lda1, 2 and 3 are delivered by *C. trachomatis* into the host cell, where they associate with LDs [6]. Lda3 also localizes at the inclusion membrane and lumen and it seems to play a major role in the recruitment of LDs by replacing adipocyte differentiation-related protein at the surface of LDs, probably facilitating the establishment of links between these organelles and the inclusion [5]. A model was previously proposed, in which via Lda3 unidentified Inc(s) (IncX) could capture LDs at the inclusion membrane, which by invagination would deliver LDs into the lumen of the inclusion [5]. Although CT006 was not previously detected in LDs isolated from *C. trachomatis*-infected cells, we speculate that CT006 could be involved in this process. In fact, different studies indicate that some Inc and non-Inc proteins interacting with LDs are being neglected. For instance, among several Incs tested, one study only detected IncA associated with LDs isolated from *C. trachomatis*-infected cells [5], while another study only detected IncG, CT618 and Cap1 [10]. However, the present and previous data does not allow to establish a possible relation between CT006, the hypothetical IncX, IncA, IncG, CT618 and Cap1.

LDs are composed of a hydrophobic core of neutral lipids surrounded by a phospholipid monolayer and associated proteins. They interact with other cellular organelles, regulate lipid and energy homeostasis [45] and are targeted by several intracellular pathogens [33, 52]. With respect to *C. trachomatis*, LDs are known to accumulate at the periphery of the inclusion and to translocate into the inclusion lumen [5, 6]. Moreover, the proteome of LDs was found to be altered during *C. trachomatis* infection [10]. It was also suggested that the absence of LDs impairs *C. trachomatis* growth [6], although this finding has been challenged in subsequent studies [7, 53]. Recently, it was shown that the depletion of specific host SNAREs increased the content of LDs in *C. trachomatis*-infected cells and decreased bacterial growth [4]. Whether the content of LDs influences or not *C. trachomatis* growth during infection of tissue cultured cells is still a matter of debate, but the relevance of LDs during chlamydial infections *in vivo* is further supported by their detection within inclusions from cells of mice infected by *Chlamydia muridarum* [9], a chlamydial species that infects rodents. Future studies combining the ability to genetically manipulate *C. trachomatis* with knowledge on the different chlamydial proteins, such as CT006, that can target LDs should eventually clarify the significance of the *Chlamydia*-LDs interaction.

## Materials and methods

### Plasmids and oligonucleotides

Plasmids and oligonucleotides used in this work are listed in S4 and S5 Tables, respectively, as well as their relevant characteristics. Plasmids were generated using restriction enzymes or by

restriction-free cloning [54]. For cloning using restriction enzymes, plasmids were constructed and purified using standard molecular biology procedures, using Phusion high-fidelity DNA polymerase (Thermo Fisher Scientific), restriction enzymes (Thermo Fisher Scientific), T4 DNA Ligase (Thermo Fisher Scientific), DreamTaq DNA polymerase (Thermo Fisher Scientific), NZYTaqII (NZYTech), DNA clean & concentrator™-5 kit, Zymoclean™ gel DNA recovery kit (Zymo Research), and GeneElute Plasmid Miniprep kit (Sigma-Aldrich) or NZYMidiprep kit (NZYTech) according to manufacturer's instructions. For restriction-free cloning, plasmids were generated using a PCR-based method [54] with Phusion high-fidelity DNA polymerase (Thermo Fisher Scientific), and DpnI (Thermo Fisher Scientific) was used to degrade parental plasmids. The accuracy of the nucleotide sequence of all the inserts in the constructed plasmids was confirmed by DNA sequencing.

For the generation of plasmid-encoded Inc-2HA or Inc-GSK fusion proteins, inserts were composed of DNA sequences comprising the *inc* gene plus 300 base pairs upstream from the transcription start codon to include the promoter region.

**Yeast strains and Vps assays.**   Yeast strains used in this work are listed in S1 Table. For Vps assays, *S. cerevisiae* NSY01 cells producing GFP or GFP-Pep12$_{L-TM}$ fusion proteins were grown in plates with yeast nitrogen base uracil dropout (YNB-Ura) supplemented with 2% (w/v) fructose at 30˚C for 3 days. Drops containing yeasts at an amount equivalent to an optical density at 600 nm (OD$_{600}$) of 0.1 in 10 μl sterile H$_2$0 were plated on the same media supplemented with 2% (w/v) fructose (non-inducing media) or 2% (w/v) galactose (inducing media) and incubated at 30˚C for 48 h. Assays for qualitative detection of invertase activity were performed as described [28].

**Cell lines and transient transfection.**   HeLa 229 and Vero cells (from the European Collection of Cell Culture; ECACC) were maintained in Dulbecco's modified Eagle Medium (DMEM; Thermo Fisher Scientific) supplemented with 10% (v/v) heat-inactivated fetal bovine serum (FBS; Thermo Fisher Scientific) at 37˚C in a 5% (v/v) CO$_2$ incubator. HeLa cells were transfected using the jetPEI™ reagent (Polyplus-Transfection) according to manufacturer's instructions. Briefly, HeLa 229 cells were seeded in 24-well plates. After 24 h, 250 ng of plasmid DNA and 1.5 μl of jetPEI™ reagent were added per well. The plate was centrifuged at 180 x *g* for 5 min at room temperature and then incubated at 37˚C in a 5% (v/v) CO$_2$ incubator. At the indicated times post-transfection, cells were collected for immunoblotting or fixed for fluorescence microscopy analysis. In the experiments where HeLa cells were transfected with plasmids and infected by *C. trachomatis*, the transfection was performed at time zero of infection.

**Bacterial strains and growth conditions.**   *Escherichia coli* NEB 10β (New England Biolabs) was used for construction and purification of plasmids, and *E. coli* ER2925 (New England Biolabs) was used to purify plasmids for transformation of *C. trachomatis*. *E. coli* strains were grown at 37˚C in liquid or solid lysogeny broth media (NZYTech) with the appropriate antibiotics and supplements.

*C. trachomatis* serovar L2 prototype strain 434/Bu (L2/434 from ATCC) was propagated in HeLa 229 cells using standard procedures [55]. Throughout this work we used the nomenclature of the annotated *C. trachomatis* D/UW3 strain [56]. *C. trachomatis* transformants were generated as described by Agaisse and Derré [57] and selected using 1 U/ml penicillin G. When transformants were observed, one passage of the bacteria was performed in the presence of 10 U/ml penicillin G, followed by clone isolation by plaque purification using Vero cells [58]. Infection procedures and quantification of infectious progeny was done as previously described [41]. At the indicated times post-infection, cells were collected and analyzed by immunoblotting or by immunofluorescence microscopy.

**Antibodies, fluorescent dyes and treatment with oleic acid.**   The following antibodies were used for immunoblotting: rabbit anti-GFP (Abcam; 1:1000), mouse anti-

phosphoglycerate kinase 1 (PGK1) (Life Technologies; 1:1000), mouse anti-chlamydial Hsp60 (A57-B9; Thermo Fisher Scientific; 1:1000), rat anti-HA (3F10; Roche; 1:1000), mouse anti-α-tubulin (clone B-5-1-2; Sigma-Aldrich; 1:1000), rabbit anti-glycogen synthase kinase-3β (anti-GSK3β) (Cell Signalling Technology; 1:5000) and rabbit anti-phospho-GSK3β (anti-pGSKβ) (Cell Signalling Technology; 1:1000), followed by anti-rabbit, anti-mouse or anti-rat horseradish peroxidase (HRP)-conjugated secondary antibodies (GE Healthcare and Jackson ImmunoResearch; 1:10000). For immunofluorescence microscopy, the following antibodies were used: rabbit anti-GFP (Abcam; 1:200), mouse anti-chlamydial Hsp60 (A57-B9; Thermo Fisher Scientific; 1:200), rat anti-HA (3F10; Roche; 1:200), goat anti-MOMP of *C. trachomatis* (Abcam; 1:200), rabbit anti-Cap1 (kindly provided by Agathe Subtil ([59]; 1:200), followed by appropriate fluorophore-conjugated anti-rabbit, anti-mouse, anti-rat or anti-goat antibodies (Jackson ImmunoResearch; 1:200). To induce synthesis of LDs, HeLa 229 cells were incubated with cell culture media containing 100 μM oleic acid (Sigma-Aldrich; stock solution at 100 mM in ethanol) for 6 hours and then fixed at the indicated time-points. For fluorescence microscopy analysis, LDs were stained with Oil Red O [Sigma-Aldrich; (3:2 v/v Oil Red stock solution diluted in water)] or with BODIPY™ 493/503 (Invitrogen; 1:200 in PBS from a saturated stock solution). To stain yeast mitochondria, yeast cells were incubated with 20 nM Mito Red (Sigma-Aldrich) diluted in YNB-Ura for 30 minutes at 30˚C (Sigma-Aldrich). To stain yeast endocytic compartments, cells were incubated for 20 minutes with 0.5 μl of FM4-64 (stock solution at 3.2 μM) diluted in yeast extract peptone dextrose (YPD) media followed by a 20 minutes chase with unlabeled YNB media.

**Fluorescence microscopy.** Transfected and/or infected HeLa 229 cells were fixed in PBS containing 4% (w/v) paraformaldehyde (PFA) for 10 min at room temperature and permeabilized with PBS containing 0.1% (w/v) Saponin (PBSS; for transfected cells) or 0.1% (v/v) Triton X-100 (PBST; for infected cells). Immunostaining was performed with antibodies diluted in PBSS or PBST containing 10% (v/v) horse serum. Cells were washed with PBS and H$_2$O before assembling the coverslips on microscopy glass slides using Aqua-poly/Mount (Polysciences). Samples were analyzed by fluorescence microscopy and images were processed and assembled using Fiji software [43].

For yeast microscopy, *S. cerevisiae* strains were grown in plates containing YNB-Ura supplemented with 2% (w/v) fructose for 3 days at 30˚C, then streaked to YNB-Ura supplemented with 2% galactose to induce protein expression and incubated for 48 h at 30˚C. Live yeast cells were visualized directly or after fluorescent staining by fluorescence microscopy.

To calculate the area of BODIPY positive LDs at *C. trachomatis* inclusions, images as those depicted in Fig 7a were randomly selected. Using Fiji software [43], the regions of interest were defined outlining individual inclusions plus 1 μm, in order to consider both LDs co-localizing with inclusions and LDs in close proximity with inclusions, because LDs do not accumulate within the inclusion lumen, indicating they are consumed after internalization [5]. Images were inverted and thresholds were applied by default settings.

## Immunoblotting

Transfected and/or infected HeLa cells were washed with PBS and detached from plates by incubation with TrypLE Express (Thermo Fisher Scientific) for 5 min at 37˚C in a 5% (v/v) CO$_2$ incubator. Cells were collected, centrifuged and washed 2 times with ice-cold PBS. Pellets were resuspended in SDS-PAGE loading buffer, boiled for 5 min at 100˚C and incubated with benzonase (Novagen) to destroy DNA and reduce the viscosity of the samples before running on SDS-PAGE.

For detection of GFP fusion proteins in *S. cerevisiae*, yeast strains were grown for 3 days at 30˚C in YNB-Ura plates supplemented with 2% (w/v) fructose and then streaked into YNB-Ura supplemented with 2% (w/v) galactose for 2 days. An amount equivalent to $OD_{600}$ 1.7 in 20 μl SDS-loading buffer was boiled for 5 minutes and run on SDS-PAGE.

In all cases, samples were separated by 12% (v/v) SDS-PAGE (or 15% for the analysis of CT006-2HA/CT006-GSK versions and for CT449-2HA) and transferred onto 0.2 μm nitrocellulose membranes (Bio-Rad) using Trans-Blot Turbo Transfer System (BioRad). Immunoblotting detection was done with SuperSignal West Pico Chemiluminescent Substrate (Thermo Fisher Scientific) or SuperSignal West Femto Maximum Sensitivity Substrate (Thermo Fisher Scientific) (as indicated in figure legends) and exposed to Amersham Hyperfilm ECL (GE Healthcare).

## Statistical analyses

Statistical analyses were done using GraphPad Prism, version 7.00 for Windows, GraphPad Software, San Diego California, USA (www.graphpad.com). For comparisons, a Shapiro-Wilk normality test was performed. As normality could not be achieved in the Shapiro-Wilk normality test, Kruskal-Wallis test was used. The α-level was set to 0.05 and a difference with $P<0.05$ was considered to be statistically significant. For comparisons between values normalized against a control group, the Wilcoxon signed-rank test was used.

## Supporting information

**S1 Fig. Analysis of the production of Inc-GFP fusion proteins in yeast by immunoblotting.** Whole cell extracts from *S. cerevisiae* NSY01 producing the indicated Inc fragments fused to GFP were analyzed by immunoblotting using antibodies against GFP and PGK1 (yeast loading control) and appropriate HRP-conjugated secondary antibodies. (a) Proteins were detected using SuperSignal West Pico detection kit (Thermo Fisher Scientific). (b) Proteins were detected using SuperSignal West Femto detection kit (Thermo Fisher Scientific). * Represents proteins that migrated according to the predicted molecular mass; # represents proteins that migrated below the predicted molecular mass; the cross in (a) corresponds to a fusion protein that was not analyzed in this study.
(PDF)

**S2 Fig. Analysis of the production of Inc-GFP-Pep12$_{L-TM}$ fusion proteins in yeast by immunoblotting.** Whole cell extracts from *S. cerevisiae* NSY01 producing the indicated Inc fragments fused to GFP-Pep12$_{L-TM}$ were analyzed by immunoblotting using antibodies against GFP and PGK1 (yeast loading control) and appropriate HRP-conjugated secondary antibodies. (a) Proteins were detected using SuperSignal West Pico detection kit (Thermo Fisher Scientific). (b) Proteins were detected using SuperSignal West Femto detection kit (Thermo Fisher Scientific). * Represents proteins that migrated according to the predicted molecular mass; # represents proteins that migrated below the predicted molecular mass.
(PDF)

**S3 Fig. The effect of Inc-GFP proteins on vacuolar protein sorting in yeast.** *S. cerevisiae* NSY01 strains producing the indicated Inc fragments fused to GFP (Inc-GFP) were grown in solid media under inducing (galactose; +GAL) or non-inducing (fructose; +FRU) conditions. After 48h, the Vps phenotype was analyzed qualitatively in solid media. Inc-GFP protein interfering with trafficking: CT229$_{91-215}$-GFP; Negative control: GFP; Positive controls: the *Legionella pneumophila* effector VipA and the dominant-negative form of the yeast ATPase Vps4

(Vps4$^{E233Q}$). Vps results with all yeast strains producing Inc-GFP proteins are summarized in S2 Table.
(PDF)

**S4 Fig. The effect of Inc-GFP-Pep12$_{L-TM}$ proteins on vacuolar protein sorting in yeast.** *S. cerevisiae* strains producing the indicated Inc fragments fused to GFP-Pep12$_{L-TM}$ (Inc-GFP-Pep12$_{L-TM}$) were grown in solid media under inducing (galactose; +GAL) or non-inducing (fructose; +FRU) conditions. After 48 h, the Vps phenotype was analyzed qualitatively in solid media. Inc-GFP-Pep12$_{L-TM}$ protein interfering with trafficking: CT223$_{192-268}$-GFP-Pep12$_{L-TM}$; Negative controls: GFP and GFP-Pep12$_{L-TM}$; Positive controls: the *Legionella pneumophila* effector VipA and the dominant-negative form of the yeast ATPase Vps4 (Vps4$^{E233Q}$). *CT135$_{1-209}$ is fused only to GFP (*CT135$_{1-209}$-GFP). Vps results with all yeast strains producing Inc-GFP-Pep12$_{L-TM}$ proteins are summarized in S3 Table.
(PDF)

**S5 Fig. Intracellular localization of Inc-GFP and Inc-GFP-Pep12$_{L-TM}$ proteins in yeast.** *S. cerevisiae* strains producing the indicated Inc-GFP proteins were grown in the presence of galactose. Live cells were visualized by fluorescence microscopy. Scale bars, 5 μm. (a) Examples for the intracellular localization of Inc-GFP proteins in yeast: cytosolic distribution (GFP and CT006$_{139-189}$-GFP), mitochondria-like puncta (CT018$_{1-90}$-GFP), endosomal compartments (CT229$_{91-215}$-GFP) and lipid droplets (CT006$_{1-88}$-GFP). (b) Examples for the intracellular localization of Inc-GFP-Pep12$_{L-TM}$ proteins at endosomal compartments in yeast. The intracellular localization of all Inc-GFP and Inc-GFP-Pep12$_{L-TM}$ fusion proteins analyzed in this study is summarized in S2 and S3 Tables.
(PDF)

**S6 Fig. Prediction of transmembrane helices in CT006.** According to TMHMM server, v 2.0 (Sonnhammer ELL, Krogh A. A hidden Markov model for predicting transmembrane helices in protein sequence. Sixth Int Conf Intell Syst Mol Biol. 1998;6:175–182; Krogh A, Larsson B, Von Heijne G, Sonnhammer ELL. Predicting transmembrane protein topology with a hidden Markov model: Application to complete genomes. J Mol Biol. 2001;305(3):567–580). CT006 is predicted to have 3 transmembrane domains between amino acid residues 47 and 69, 89 and 111, and 118 and 140. The positions of the last two hydrophobic motifs are approximately in agreement with the prediction of the bilobed hydrophobic motif described by Dehoux *et al*, 2011 (Dehoux P, Flores R, Dauga C, Zhong G, Subtil A. Multi-genome identification and characterization of chlamydiae-specific type III secretion substrates: The Inc proteins. BMC Genomics. 2011;12:109).
(PDF)

**S7 Fig. Analysis of the production and intracellular localization of CT006 versions in mammalian cells.** HeLa 229 cells were transfected for 24 h with plasmids encoding mEGFP or the indicated versions of CT006 containing a mEGFP tag at their amino-termini (mEGFP-CT006 proteins) or at their carboxy-termini (CT006-mEGFP proteins). (a) Transfected cells were fixed with 4% (w/v) PFA and imaged by fluorescence microscopy. Scale bars, 10 μm. (b) Whole cell extracts were analyzed by immunoblotting with antibodies against GFP and α-tubulin (HeLa 229 cells loading control) and appropriate HRP-conjugated secondary antibodies. Proteins were detected using SuperSignal West Pico detection kit (Thermo Fisher Scientific). The crosses in (a) correspond to proteins that were not analyzed in this study.
(PDF)

**S8 Fig. Full-length CT006 and CT006$_{1-88}$ fused to mEGFP partially co-localize with the endoplasmic reticulum in mammalian cells.** HeLa 229 cells were transfected with plasmids encoding mEGFP or the indicated versions of CT006 containing a mEGFP tag at their amino-termini (mEGFP-CT006 proteins). After 18 h, cells were treated ethanol (solvent control; left-hand side panel) or 100 μM oleic acid (right-hand side panel) for 6 h and fixed with 4% (w/v) PFA. Fixed cells were immunolabeled with an antibody against Protein disulfide isomerase (PDI), and an appropriate fluorophore-conjugated secondary antibody, and imaged by fluorescence microscopy. Scale bars, 10 μm.
(PDF)

**S9 Fig. Analysis of the production of mEGFP-CT006$_{1-88}$ versions with positively charged amino acids replaced by glycines in mammalian HeLa cells.** HeLa 229 cells were transfected for 24 h with plasmids encoding mEGFP or the indicated versions of CT006$_{1-88}$ containing a mEGFP tag at their amino-termini. Whole cell extracts were analyzed by immunoblotting with antibodies against GFP and α-tubulin (HeLa 229 cells loading control) and appropriate HRP-conjugated secondary antibodies. Proteins were detected using SuperSignal West Pico detection kit (Thermo Fisher Scientific).
(PDF)

**S10 Fig. Plasmid-encoded CT006-2HA is produced at early times post-infection and accumulates at the periphery of the inclusion.** HeLa 229 cells were infected by *C. trachomatis* L2/434 or *C. trachomatis* L2/434 harboring pCT006-2HA (L2/434+pCT006-2HA). At the indicated hours post-infection (hpi), cells were fixed with 4% (w/v) PFA, immunolabeled with antibodies against HA (red), *C. trachomatis* Hsp60 (green), and appropriate fluorophore-conjugated secondary antibodies, and imaged by fluorescence microscopy. Scale bars, 10 μm. Dashed lines represent the limits of infected HeLa cells and the chlamydial inclusions within were zoomed at 2, 4, 6 and 8 hpi.
(PDF)

**S11 Fig. mEGFP-CT449 does not localize at lipid droplets in mammalian cells.** HeLa 229 cells were transfected with plasmids encoding mEGFP or different versions of CT449 containing a mEGFP tag at their amino-termini (mEGFP-CT449$_{FL}$, mEGFP-CT449$_{1-41}$ or mEGFP-CT449$_{88-110}$). (a) At 24 h post-transfection, whole cell extracts were analyzed by immunoblotting with antibodies against GFP and α-tubulin (HeLa 229 cells loading control) and appropriate HRP-conjugated secondary antibodies. (b) At 24 h post-transfection, cells were fixed with 4% (w/v) PFA and analyzed by fluorescence microscopy. (c) At 18 h post-transfection, cells were treated with 100 μM oleic acid for 6 h and then fixed with 4% (w/v) PFA. Fixed cells were labeled with anti-GFP and the appropriate fluorophore-conjugated secondary antibody, stained with Oil Red O (3:2 v/v Oil Red O stock solution diluted in water), and imaged by fluorescence microscopy. Scale bars, 10 μm. The area delimited by a white square was zoomed.
(PDF)

**S12 Fig. Analysis of the production of plasmid-encoded CT449-2HA, CT006$_{5G}$-2HA and CT006$_{Δ47-67}$-2HA by *C. trachomatis*.** HeLa 229 cells were left uninfected (UI) or infected by *C. trachomatis* L2/434 or by L2/434 strains harboring (a) pCT449-2HA, (b) pCT006$_{5G}$-2HA, or (c) pCT006$_{Δ47-67}$-2HA. At the indicated times post-infection, whole cell extracts were analyzed by immunoblotting using antibodies against HA, Hsp60 (bacterial loading control) and α-tubulin (HeLa 229 cells loading control) and the appropriate HRP-conjugated secondary antibodies, followed by detection using SuperSignal West Pico detection kit (Thermo Fisher Scientific).
(PDF)

**S13 Fig. Analysis of the intracellular localization of plasmid-encoded CT006-2HA, CT449-2HA, CT006$_{5G}$-2HA and CT006$_{\Delta47-67}$-2HA.** HeLa 229 cells were infected by L2/434 strains harboring pCT006-2HA, pCT449-2HA, pCT006$_{5G}$-2HA or pCT006$_{\Delta47-67}$-2HA. At the indicated times post-infection, infected cells were fixed with 4% (w/v) PFA, immunolabeled with antibodies against HA (red), Hsp60 (green) and appropriate fluorophore-conjugated secondary antibodies. The labeled cells were then imaged by fluorescence microscopy. Scale bars, 10 µm.
(PDF)

**S14 Fig. Analysis of the localization of LDs in cells infected by *C. trachomatis*.** HeLa 229 cells were infected by *C. trachomatis* L2/434 strains harboring pCT449-2HA, pCT006$_{5G}$-2HA or pCT006$_{\Delta47-67}$-2HA. At the indicated times post-infection, infected cells previously treated with 100 µM oleic acid for 6 h were fixed with 4% (w/v) PFA, immunolabeled with an antibody against Hsp60 (red) and an appropriate fluorophore-conjugated secondary antibody, stained with the neutral lipid dye BODIPY (green) and imaged by fluorescence microscopy. Dashed lines represent the limits of infected HeLa 229 cells. Scale bars, 10 µm.
(PDF)

**S15 Fig. Analysis of the area of LDs at the region of inclusions.** (a) HeLa 229 cells were infected by the *C. trachomatis* L2/434 parental strain or by *C. trachomatis* L2/434 strains harboring pCT006-2HA (L2/434+pCT006-2HA) or pCT449-2HA (L2/434+pCT449-2HA). (b) HeLa 229 cells were infected by the *C. trachomatis* L2/434 parental strain or by *C. trachomatis* L2/434 strains harboring pCT006-2HA (L2/434+pCT006-2HA), pCT006$_{5G}$-2HA (L2/434 +pCT006$_{5G}$-2HA) or pCT449-2HA (L2/434+pCT449-2HA). The area of BODIPY-positive LDs at the region of chlamydial inclusions was measured from randomly selected images, as those depicted in Fig 7a and S14 Fig, using Fiji software (Schindelin J, Arganda-Carreras I, Frise E, Kaynig V, Longair M, Pietzsch T, *et al.* Fiji: An open-source platform for biological-image analysis. Nat Methods. 2012;9(7):676–682). Data are represented as the mean and standard error of the mean from three independent experiments (n = 60). Statistical analysis was performed using Kruskal-Wallis test with Dunn's multiple comparison test. * Represents P<0.05 by comparison to the L2/434 strain; ϕ represents P<0.05 by comparison to the L2/434 +pCT006-2HA.
(PDF)

**S16 Fig. Uncropped blot images.** Uncroppped blot images from Figs 5a and 6a, Fig a in S1 Fig, Fig b in S1 Fig, Fig a in S2 Fig, Fig b in S2 Fig, Fig b in S7 Fig, S9 Fig, Fig a in S11 Fig, Fig a in S12 Fig, Fig b in S12 Fig, and Fig c in S12 Fig.
(PDF)

**S1 Table. *Saccharomyces cerevisiae* strains used in this work.**
(PDF)

**S2 Table. Inc-GFP fusion proteins—Summary of production and localization in *Saccharomyces cerevisiae*, and induction of a vacuolar protein sorting (Vps) defect.**
(PDF)

**S3 Table. Inc-GFP-Pep12$_{L-TM}$ fusion proteins—Summary of production and localization in *Saccharomyces cerevisiae*, and induction of a vacuolar protein sorting (Vps) defect.**
(PDF)

**S4 Table. Plasmids used in this work.**
(PDF)

**S5 Table. Oligonucleotides used in this work.**
(PDF)

## Acknowledgments

We thank Irina Franco for critical reading of the manuscript.

## Author Contributions

**Conceptualization:** Luís Jaime Mota.

**Formal analysis:** Joana N. Bugalhão, Luís Jaime Mota.

**Investigation:** Joana N. Bugalhão, Maria P. Luís, Inês S. Pereira, Maria da Cunha, Sara V. Pais.

**Methodology:** Joana N. Bugalhão, Luís Jaime Mota.

**Project administration:** Luís Jaime Mota.

**Resources:** Joana N. Bugalhão, Maria P. Luís, Inês S. Pereira, Maria da Cunha, Sara V. Pais.

**Supervision:** Luís Jaime Mota.

**Writing – original draft:** Joana N. Bugalhão, Luís Jaime Mota.

**Writing – review & editing:** Joana N. Bugalhão, Maria P. Luís, Inês S. Pereira, Maria da Cunha, Sara V. Pais, Luís Jaime Mota.

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
