## [Decision Letter · Decision Letter 0]

5 Jan 2022

PONE-D-21-38824The Chlamydia trachomatis inclusion membrane protein L (IncL) associates with host cell lipid dropletsPLOS ONE

Dear Dr. Mota,

Thank you for submitting your manuscript to PLOS ONE. After careful consideration, we feel that it has merit but does not fully meet PLOS ONE’s publication criteria as it currently stands. Therefore, we invite you to submit a revised version of the manuscript that addresses the points raised during the review process. 

While both reviewers thought the experiments were well done, they also expressed concerns that the data does not support the overall conclusion that CT006 associates with lipid droplets during Chlamydia infection.

We look forward to receiving your revised manuscript.

Kind regards,

Stacey D Gilk, Ph.D.

Academic Editor

PLOS ONE

Journal Requirements:

"We thank Irina Franco for critical reading of the manuscript. This work was supported by Fundação para a Ciência e Tecnologia (FCT) through grants PTDC/BIA-MIC/28503/2017 and PTDC/IMI-MIC/1300/2014, and in the scope of the projects UIDP/04378/2020 and UIDB/04378/2020 of the Research Unit on Applied Molecular Biosciences – UCIBIO, and LA/P/0140/2020 of the Associate Laboratory Institute for Health and Bioeconomy - i4HB. JNB and SVP were supported by PhD fellowships PD/BD/128214/2016 and PD/BD/52210/2013, respectively, within the scope of the PhD program Molecular Biosciences (PD/00133/2012) funded by FCT. ISP and MPL were supported by PhD fellowships SFRH/BD/129756/2017 and SFRH/BD/144284/2019, also funded by FCT."

"This work was supported by Fundação para a Ciência e Tecnologia (FCT) through grants PTDC/BIA-MIC/28503/2017 and PTDC/IMI-MIC/1300/2014 attributed to LJM, and in the scope of the projects UIDP/04378/2020 and UIDB/04378/2020 of the Research Unit on Applied Molecular Biosciences – UCIBIO, and LA/P/0140/2020 of the Associate Laboratory Institute for Health and Bioeconomy - i4HB. JNB and SVP were supported by PhD fellowships PD/BD/128214/2016 and PD/BD/52210/2013, respectively, within the scope of the PhD program Molecular Biosciences (PD/00133/2012) funded by FCT. ISP and MPL were supported by PhD fellowships SFRH/BD/129756/2017 and SFRH/BD/144284/2019, also funded by FCT. The funders had no role in study design, data collection and analysis, decision to publish, or preparation of the manuscript."

Reviewers' comments:

Reviewer's Responses to Questions

**Comments to the Author**

1. Is the manuscript technically sound, and do the data support the conclusions?

Reviewer #1: Yes

Reviewer #2: No

2. Has the statistical analysis been performed appropriately and rigorously? 

Reviewer #1: Yes

Reviewer #2: Yes

3. Have the authors made all data underlying the findings in their manuscript fully available?

Reviewer #1: Yes

Reviewer #2: Yes

4. Is the manuscript presented in an intelligible fashion and written in standard English?

Reviewer #1: Yes

Reviewer #2: Yes

5. Review Comments to the Author

Reviewer #1: In this manuscript by Bugalhão et al., the authors used several large-scale screens to preliminarily characterize Chlamydia trachomatis inclusion membrane proteins (Incs). They demonstrate that 2 Incs perturb host vesicular trafficking, which is largely in support of previous studies. Using ectopic expression, they demonstrate that several Incs traffic to eukaryotic organelles. Uniquely, they demonstrate that CT006 (IncL) might target lipid droplets. Overall, this is a nice descriptive study that provides some insight into the putative function of CT006. My comments are minor, namely a few missing citations and that it is premature to rename CT006 to IncL. I have included specifics below.

Missing citations:

Mital J (2013) PLOS One- Ectopically expressed several Incs

Weber MM (2015) Infect Immun- demonstrated that CT006 is a bona fide Inc

Based on the data presented herein, I believe it is premature to rename CT006 to IncL. The observations that it might target lipid droplets largely relies on ectopic data and minimal infection experiments. Without a mutant to confirm it is important for lipid droplet recruitment or a binding partner, it is best to keep its designation as CT006.

Reviewer #2: Intravacuolar Chlamydia trachomatis is notorious for interacting with many host mammalian organelles. Identifying bacterial effectors (eg., Inc proteins) exported at the inclusion membrane or secreted into the host cell, that mediate these interactions are important for better understanding the chlamydial pathogenicity. To identify novel C. trachomatis Incs interfering with host vesicular trafficking, thus potentially interacting with host vesicles/organelles, the authors have undertaken a comprehensive approach: they used S. cerevisiae to ectopically express cytosolic domains of Incs to monitor vacuolar protein sorting mistrafficking in yeast. They identified CT223 and CT229 capable of causing sorting defects in yeast. The paper focuses on CT229 localization.

Overall, the assays performed in the manuscript are well-done (no technical flaws) but the interpretation of the results is erroneous and misleading as we cannot conclude from the data that CT229 (renamed IncL) associates with host cell lipid droplets (LD) for several reasons:

1. the authors have expressed CT229 FL or fragments in yeast and mammalian cells: CT229 FL is cytosolic, CT22991-215 is endosomal and CT2291-88 is targeted to LD. Thus positively charged sequences close to the amino-terminal hydrophobic domain CT229 (between aa 1 and 88) are likely responsible for the tropism of this peptide to LD – same situation with positively charged sequences of caveolin that mediate the sorting of caveolin to LD while central hydrophobic domain anchors of caveolin direct the protein to the ER. However, we cannot extrapolate from these data that CT229 physiologically interact with host LD in infected cells, more especially if CT229 FL does not localize to LD in yeast and mammalian cells.

2. The localization of CT229 assessed by IFA in Chlamydia strains expressing IncL-2HA (with anti-HA) or GSK fused to IncL (with anti-GSK) is at the inclusion membrane from 16 to 24hpi (Fig. 6). In the attempt to examine whether CT229 is at least secreted into the host cell for LD interaction, the authors looked at CT229 localization at early time points (2 to 8hpi in Fig. S10), and observed a signal beyond the cytosolic signal of Hsp60. They conclude that CT229 is on extensions of the inclusion membrane. However, without showing a colocalization of CT229 with IncA or IncG that are expressed on extensions of the inclusion membrane, we cannot ascertain that CT229 are on extensions of the inclusion membrane. Irrespective to these data, CT229 is not observed close/around host LD: in Fig. 6. the CT229 red signal from IncL-2HA does not intersect with the green LD signal. The only conclusion of Fig. 6 is that few host LD are in proximity to the inclusion, at the limit of the fluorescence microscopy resolution (~200nm).

3. The Discussion contains too many repetitions of the Results section (to be combined). What must be better discussed is the connection between CT229 with IncA (according to ref. 5, only IncA associate with purified host LD), IncX (in the proposed model in ref. 5) as well with IncG, CT618 and Cap1 (from ref. 10) in case of physiological role of CT229 on host LD.

6. PLOS authors have the option to publish the peer review history of their article (what does this mean?). If published, this will include your full peer review and any attached files.

Reviewer #1: No

Reviewer #2: No

---

## [Author Response · Author response to Decision Letter 0]

17 Jan 2022

Academic Editor

1 - “While both reviewers thought the experiments were well done, they also expressed concerns that the data does not support the overall conclusion that CT006 associates with lipid droplets during Chlamydia infection.”

Response: We agree that our data does not allow us to conclude that CT006 associates with lipid droplets during Chlamydia infection. To clarify this, we modified the title to “The Chlamydia trachomatis inclusion membrane CT006 associates with lipid droplets in eukaryotic cells” as well as in the remainder of the text whenever the writing could be ambiguous in this regard. 

Reviewer #1

1 - “Missing citations:

Mital J (2013) PLOS One- Ectopically expressed several Incs

Weber MM (2015) Infect Immun- demonstrated that CT006 is a bona fide Inc”

Response: These papers are now cited. 

2 - “Based on the data presented herein, I believe it is premature to rename CT006 to IncL. The observations that it might target lipid droplets largely relies on ectopic data and minimal infection experiments. Without a mutant to confirm it is important for lipid droplet recruitment or a binding partner, it is best to keep its designation as CT006.”

Response: we modified the manuscript throughout (text, figures, and supplementary data) replacing IncL by CT006.

Reviewer #2

1 – “the authors have expressed CT229 FL or fragments in yeast and mammalian cells: CT229 FL is cytosolic, CT22991-215 is endosomal and CT2291-88 is targeted to LD. Thus positively charged sequences close to the amino-terminal hydrophobic domain CT229 (between aa 1 and 88) are likely responsible for the tropism of this peptide to LD – same situation with positively charged sequences of caveolin that mediate the sorting of caveolin to LD while central hydrophobic domain anchors of caveolin direct the protein to the ER. However, we cannot extrapolate from these data that CT229 physiologically interact with host LD in infected cells, more especially if CT229 FL does not localize to LD in yeast and mammalian cells.”

Response: We agree that our data does not allow to extrapolate that CT006 (the protein in which we focused our studies) interacts with host LDs in infected cells. Please see Response to the Academic Editor.

2 – “The localization of CT229 assessed by IFA in Chlamydia strains expressing IncL-2HA (with anti-HA) or GSK fused to IncL (with anti-GSK) is at the inclusion membrane from 16 to 24hpi (Fig. 6). In the attempt to examine whether CT229 is at least secreted into the host cell for LD interaction, the authors looked at CT229 localization at early time points (2 to 8hpi in Fig. S10), and observed a signal beyond the cytosolic signal of Hsp60. They conclude that CT229 is on extensions of the inclusion membrane. However, without showing a colocalization of CT229 with IncA or IncG that are expressed on extensions of the inclusion membrane, we cannot ascertain that CT229 are on extensions of the inclusion membrane. Irrespective to these data, CT229 is not observed close/around host LD: in Fig. 6. the CT229 red signal from IncL-2HA does not intersect with the green LD signal. The only conclusion of Fig. 6 is that few host LD are in proximity to the inclusion, at the limit of the fluorescence microscopy resolution (~200nm).”

Response: We previously wrote “At 8 h post-infection IncL-2HA appeared in small tubules, which are likely an extension of the vacuolar membrane”. To address the issue raised by the reviewer that “we cannot ascertain that CT229 are on extensions of the inclusion membrane”, and as we do not have anti-IncA or anti-IncG antibodies, we now modified the text to “At 8 h post-infection CT006-2HA appeared in small tubules near the inclusion, suggesting that these tubules could be an extension of the vacuolar membrane”. Being on the inclusion membrane, CT006, as any other Inc, can potentially interact with LDs on the host cell cytoplasm. We did not analyse a co-localization between CT006 and LDs by microscopy (our Fig. 6 shows CT006, Cap1 and chlamydial MOMP) because as CT006 is apparently homogenously distributed around the inclusion a normal overlap with some LDs would have little significance.

3 – “The Discussion contains too many repetitions of the Results section (to be combined). What must be better discussed is the connection between CT229 with IncA (according to ref. 5, only IncA associate with purified host LD), IncX (in the proposed model in ref. 5) as well with IncG, CT618 and Cap1 (from ref. 10) in case of physiological role of CT229 on host LD.”

Response: The discussion was abridged (~500 words) to avoid repetitions of the Results section, but we do prefer to keep the Discussion and Results as separate sections. We previously mentioned IncX (while not explicitly), and the previously shown association of IncA (in one study) and of IncG, CT618 and Cap1 (in another study) with LDs. From the existing data (ours and from these publications) we think the connection between CT006 and these proteins is not obvious and this is now clearly written.

---

## [Decision Letter · Decision Letter 1]

8 Feb 2022

The Chlamydia trachomatis inclusion membrane protein CT006 associates with lipid droplets in eukaryotic cells

PONE-D-21-38824R1

Dear Dr. Mota,

We’re pleased to inform you that your manuscript has been judged scientifically suitable for publication and will be formally accepted for publication once it meets all outstanding technical requirements.

Kind regards,

Stacey D Gilk, Ph.D.

Academic Editor

PLOS ONE

Additional Editor Comments (optional):

Reviewers' comments:

Reviewer's Responses to Questions

**Comments to the Author**

1. If the authors have adequately addressed your comments raised in a previous round of review and you feel that this manuscript is now acceptable for publication, you may indicate that here to bypass the “Comments to the Author” section, enter your conflict of interest statement in the “Confidential to Editor” section, and submit your "Accept" recommendation.

Reviewer #1: All comments have been addressed

Reviewer #2: All comments have been addressed

2. Is the manuscript technically sound, and do the data support the conclusions?

Reviewer #1: Yes

Reviewer #2: Yes

3. Has the statistical analysis been performed appropriately and rigorously? 

Reviewer #1: Yes

Reviewer #2: Yes

4. Have the authors made all data underlying the findings in their manuscript fully available?

Reviewer #1: Yes

Reviewer #2: Yes

5. Is the manuscript presented in an intelligible fashion and written in standard English?

Reviewer #1: Yes

Reviewer #2: Yes

6. Review Comments to the Author

Reviewer #1: (No Response)

Reviewer #2: It is indeed safer to keep CT6006 and avoid using IncL at this stage.

For future work on host organelle-inclusion interaction, it will be better to have an antibody against the inclusion membrane. Many labs have some, just ask.

7. PLOS authors have the option to publish the peer review history of their article (what does this mean?). If published, this will include your full peer review and any attached files.

Reviewer #1: No

Reviewer #2: No

---

## [Editor Report · Acceptance letter]

11 Feb 2022

PONE-D-21-38824R1 

The *Chlamydia trachomatis* inclusion membrane protein CT006 associates with lipid droplets in eukaryotic cells 

Dear Dr. Mota:

I'm pleased to inform you that your manuscript has been deemed suitable for publication in PLOS ONE. Congratulations! Your manuscript is now with our production department. 

Kind regards, 

on behalf of

Dr. Stacey D Gilk 

Academic Editor

PLOS ONE